EMBO
Molecular Medicine

# *NRAS* destines tumor cells to the lungs

Anastasios D Giannou[1,†], Antonia Marazioti[1,†], Nikolaos I Kanellakis[1,†] (iD), Ioanna Giopanou[1,†],
Ioannis Lilis[1], Dimitra E Zazara[1], Giannoula Ntaliarda[1], Danai Kati[1], Vasileios Armenis[1],
Georgia A Giotopoulou[1], Anthi C Krontira[1], Marina Lianou[1], Theodora Agalioti[1], Malamati Vreka[1,2],
Maria Papageorgopoulou[1], Sotirios Fouzas[3], Dimitrios Kardamakis[4], Ioannis Psallidas[1,5,‡],
Magda Spella[1,*,‡] (iD) & Georgios T Stathopoulos[1,2,‡,**] (iD)

## Abstract

The lungs are frequently affected by cancer metastasis. Although *NRAS* mutations have been associated with metastatic potential, their exact role in lung homing is incompletely understood. We cross-examined the genotype of various tumor cells with their ability for automatic pulmonary dissemination, modulated NRAS expression using RNA interference and *NRAS* overexpression, identified NRAS signaling partners by microarray, and validated them using *Cxcr1-* and *Cxcr2*-deficient mice. Mouse models of spontaneous lung metastasis revealed that mutant or overexpressed *NRAS* promotes lung colonization by regulating interleukin-8-related chemokine expression, thereby initiating interactions between tumor cells, the pulmonary vasculature, and myeloid cells. Our results support a model where *NRAS*-mutant, chemokine-expressing circulating tumor cells target the CXCR1-expressing lung vasculature and recruit CXCR2-expressing myeloid cells to initiate metastasis. We further describe a clinically relevant approach to prevent NRAS-driven pulmonary metastasis by inhibiting chemokine signaling. In conclusion, NRAS promotes the colonization of the lungs by various tumor types in mouse models. IL-8-related chemokines, NRAS signaling partners in this process, may constitute an important therapeutic target against pulmonary involvement by cancers of other organs.

**Keywords** inflammation and cancer; interleukin-8-related chemokines; lung endothelium; myeloid cells; pulmonary metastasis
**Subject Categories** Cancer; Respiratory System

## Introduction

Metastasis is a central hallmark of cancer and the most common mode of cancer death (Nguyen *et al*, 2009; Vanharanta & Massague, 2013). Together with the bones, liver, and brain, the lungs are common target organs of metastasis, being affected in 25–40% of cancer patients overall and in even higher proportions of patients with some tumors (i.e. lung, breast, colon, and melanomatous skin cancers) that display predilection for the lungs (Hess *et al*, 2006; Disibio & French, 2008). Hence preventing or curing lung metastasis would likely lead to reductions in cancer deaths. For this, the mechanisms of multiple steps of the metastatic cascade need to be better understood: the mobilization of malignant cells from the primary tumor and of immune cells from the bone barrow; the homing of these cells to target organs; and the development of clinically evident metastases with structural and functional disruption (Kim *et al*, 2009; Nguyen *et al*, 2009; Borovski *et al*, 2011; Comen *et al*, 2011).

Mounting evidence indicates that metastasis cannot be explained by anatomic and physiologic factors alone and suggests the existence of metastatic traits in tumor cells (Klein, 2003; Edlund *et al*, 2004; Meuwissen & Berns, 2005; Nguyen *et al*, 2009). Indeed, multiple investigations support this hypothesis via the identification of gene signatures that drive tropism of a given cancer for a given organ (Gupta *et al*, 2007; Minn *et al*, 2007; Padua *et al*, 2008; Chen *et al*, 2011; Oskarsson *et al*, 2011; Acharyya *et al*, 2012). However, these signatures are hard to target, and direct links between a single cancer mutation and metastatic tropism to a given organ, such as those suggested by observational studies of *NRAS* and *KRAS* mutations in pulmonary and hepatic metastasis of colon cancer and melanoma (Tie *et al*, 2011; Urosevic *et al*, 2014; Lan *et al*, 2015; Pereira *et al*, 2015; Ulivieri *et al*, 2015), are intriguing and of potential clinical value.

1 Laboratory for Molecular Respiratory Carcinogenesis, Department of Physiology, Faculty of Medicine, University of Patras, Rio, Greece
2 Comprehensive Pneumology Center (CPC) and Institute for Lung Biology and Disease (iLBD), Member of the German Center for Lung Research (DZL), University Hospital, Ludwig-Maximilians University and Helmholtz Center Munich, Munich, Germany
3 Pneumology Unit, Department of Pediatrics, Faculty of Medicine, University of Patras, Rio, Greece
4 Department of Radiation Oncology and Stereotactic Radiotherapy, Faculty of Medicine, University of Patras, Rio, Greece
5 Oxford Centre for Respiratory Medicine, Oxford University Hospitals NHS Trust, Oxford, UK
*Corresponding author. Tel: +30 2610 969154; Fax: +30 2610 969176; E-mail: magsp@upatras.gr
**Corresponding author. Tel: +49 (89) 3187 4846; Fax: +49 (89) 3187 4661; E-mails: gstathop@upatras.gr; stathopoulos@helmholtz-muenchen.de
†These authors contributed equally to this work as first authors
‡These authors contributed equally to this work as senior authors

Mutations and overexpression of the neuroblastoma RAS viral (v-ras) oncogene homolog (NRAS) are found across multiple tumor types and are common in highly metastatic cancers such as tumors of unknown primary, melanomas, and sarcomas that display a striking propensity for lung metastasis (Disibio & French, 2008; Stephen et al, 2014; Forbes et al, 2015). Clinical studies suggest that patients with NRAS-mutant or amplified tumors suffer from more aggressive disease than patients with wild-type NRAS alleles (Jakob et al, 2012; Jang et al, 2014; Ulivieri et al, 2015), and one recent study identified increased frequency of lung metastases in NRAS-mutant patients (Lan et al, 2015). However, the role of the oncogene in pulmonary metastasis has not been functionally studied.

We discovered that tumor cells of various tissues of origin that carry NRAS mutations are able to spontaneously metastasize to the lungs of mice from subcutaneous (s.c.) primary sites, while cancer cells with wild-type NRAS cannot. We document that mutant or overexpressed NRAS is required for this capability of tumor cells and that it suffices to transmit it to cancer cells without NRAS mutations or even to benign cells. Importantly, we show that this phenotype of cancer cells that is triggered by NRAS is not due to enhanced growth capacities conferred by the oncogene, but rests on inflammatory chemokine signaling to cognate receptors on host lung endothelial and myeloid cells and can thus be targeted by chemokine receptor inhibition.

## Results

### An inflammatory link between NRAS and pulmonary metastasis

We initially cross-examined the genetic alterations of eleven murine and human tumor cell lines with their spontaneous growth and dissemination patterns. For this, mouse cellular RNA was Sanger-sequenced for eight common cancer genes and human cell line data were obtained from the catalogue of somatic mutations in cancer (COSMIC) cell lines project (http://cancer.sanger.ac.uk/cancergenome/projects/cell_lines/) (Ikediobi et al, 2006; Forbes et al, 2015). In parallel, tumor cells of various tissues of origin were implanted s.c. into appropriate hosts at a titer yielding 100% tumor incidence ($0.5 \times 10^6$ mouse and $10^6$ human cells) and mice were sacrificed when moribund for lung examination. Three NRAS-mutant (NRAS$^{MUT}$; Lewis lung carcinoma, LLC, and AE17 malignant pleural mesothelioma of C57BL/6 mice carried Nras$^{Q61H}$; human SKMEL2 skin melanoma carried NRAS$^{Q61R}$) and eight NRAS-wild-type (NRAS$^{WT}$; MC38 colon adenocarcinoma, B16F10 skin melanoma, and PANO2 pancreatic adenocarcinoma of C57BL/6 mice; CT26 colon adenocarcinoma and AB2 malignant pleural mesothelioma of BALB/c mice; as well as human A549 and HCC-827 lung adenocarcinomas and MDA-MB-231 breast carcinoma) tumor cell lines were identified (Fig EV1A). Six cell lines harbored KRAS, four TP53, two EGFR, one BRAF, and one STK11 mutations that coexisted with NRAS$^{MUT}$ in a random fashion (Fig 1A). While all cell lines caused flank tumors in all mice injected (155/155; Fig EV1B), they exhibited a dichotomous capacity for automatic lung metastasis: Most mice (43/46) that received NRAS$^{MUT}$ cell lines developed more than two bulky pulmonary macrometastases (diameter > 200 μm; clearly visible by the naked eye), as opposed to only 3/109 mice with NRAS$^{WT}$ tumor cells, with the remaining exhibiting only tumor

emboli and micrometastases (diameter < 200 μm; not visible by the unaided eye; Fig 1B and C). This phenotype co-segregated with NRAS, but not with KRAS mutation status or tissue of origin (Fig 1D and E). NRAS$^{MUT}$ cells also displayed overexpression of the oncogene and activation of various downstream signaling pathways (Fig EV1C and E), in accord with what is observed in human KRAS- and NRAS-mutant tumors (Stephen et al, 2014; Pfarr et al, 2016). We next transfected C56BL/6 mouse tumor cells carrying either NRAS$^{MUT}$ (LLC and AE17; Nras$^{Q61H}$) or NRAS$^{WT}$ (MC38) with a home-made plasmid encoding firefly luciferase (pCAG.Luc) to noninvasively track them in vivo after s.c. injection to syngeneic C57BL/6 hosts. All mice developed primary tumors emitting comparable bioluminescent signals that were excised after 2 weeks, but only mice with NRAS$^{MUT}$ s.c. tumor cells developed bioluminescent lung metastases after additional 2 weeks (Figs 2A and EV2A). In complementary experiments, C57BL/6 recipients were lethally irradiated, reconstituted with bioluminescent bone marrow transplants (BMT) from syngeneic CAG.Luc.eGFP mice (Cao et al, 2004; Marazioti et al, 2013; Giannou et al, 2015), and received non-luminescent tumor cells s.c. After 4 weeks, bioluminescent myeloid cells were detected in all primary tumors, but only in the thoraces of mice with NRAS$^{MUT}$ primary tumors and lung metastases (Figs 2B and EV2B). To study early premetastatic niches in the lungs, BMT was also performed using red-fluorescent mT/mG donors (Muzumdar et al, 2007) and s.c. injections of tumor cells harboring NRAS$^{MUT}$ (LLC, AE17) or NRAS$^{WT}$ (MC38) labeled using lentiviral eGFP plasmid (peGFP). In animals terminated after 2 weeks, that is, prior to frank metastasis, mononuclear and polymorphonuclear mT$^+$ myeloid cells co-segregated in lung niches with GFP$^+$ metastatic LLC and AE17 cells; in contrast, mice with s.c. MC38 tumors displayed no pulmonary GFP$^+$ tumor cells and evenly dispersed mononuclear mT$^+$ myeloid cells (Figs 2C and EV2C). Flow cytometric analysis of naïve and tumor-bearing C57BL/6 mice revealed increased numbers of circulating and splenic myeloid cells in all tumor-bearing mice, but increased pulmonary myeloid cells exclusively in the lungs of mice carrying NRAS$^{MUT}$ LLC and AE17 tumors (Figs 2D and EV2D). Taken together with published work (Lyden et al, 2001; Rafii et al, 2002; Kaplan et al, 2005; Yang et al, 2008; Chen et al, 2011; Acharyya et al, 2012), these data suggested that tumor cells with mutant NRAS possess enhanced capability for automatic metastasis to the lungs, being thereby accompanied by myeloid cells to form metastatic niches.

### NRAS drives circulating tumor cells to the lungs

We next tested whether NRAS mutation and overexpression are functionally involved in pulmonary metastasis and at which step: primary tumor escape or lung homing? For this, peGFP-labeled mouse tumor cells with NRAS$^{MUT}$ (LLC, AE17) or NRAS$^{WT}$ (MC38) were implanted s.c. in C57BL/6 mice and were chased in the blood and the lungs by flow cytometry. Interestingly, all tumor types gave rise to similar numbers of circulating tumor cells, but only NRAS$^{MUT}$ LLC and AE17 cells could home to the lungs (Figs 3A and EV2E). In addition, stable transfection of NRAS$^{WT}$ MC38 cells with NRAS$^{MUT}$-encoding vector (pNRAS$^{G61K}$; Khosravi-Far et al, 1996) and also with NRAS$^{WT}$-encoding vector (pNRAS$^{WT}$; Fiordalisi et al, 2001), but not a control (pC; Morgenstern & Land, 1990) vector, rendered them capable of

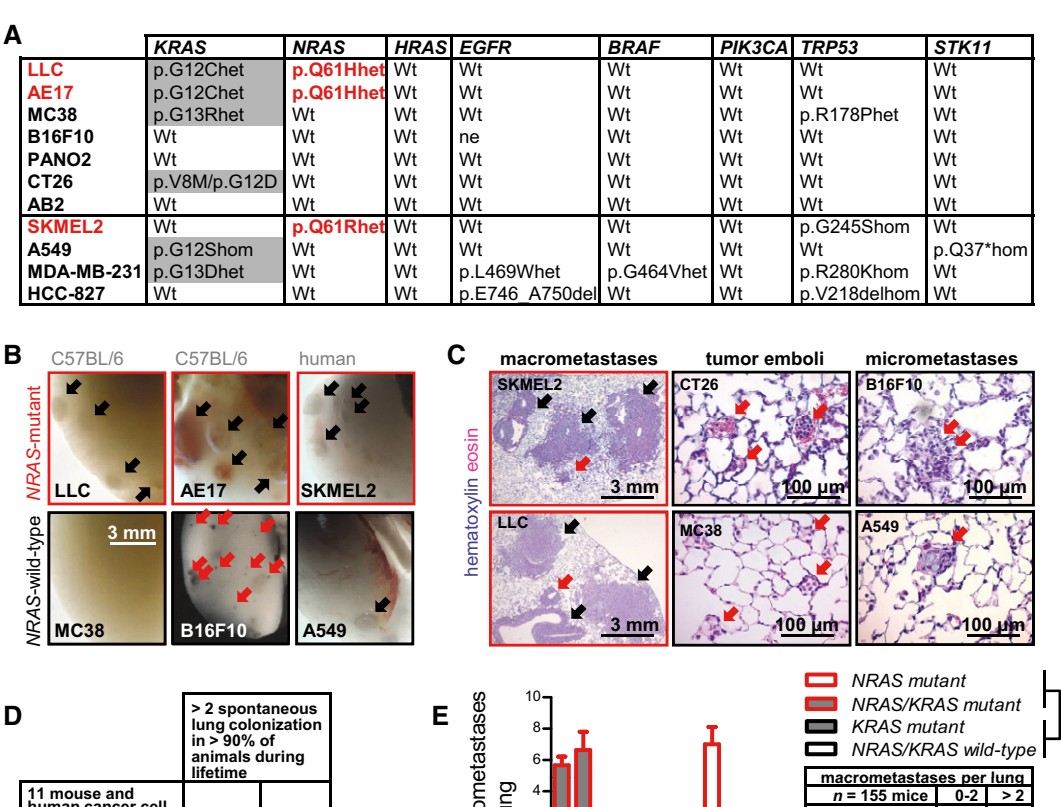

**Figure 1. NRAS mutations and spontaneous lung metastasis of mouse and human cancer cell lines.**

A    Mutation summary of eight cancer genes sequenced in seven mouse cancer cell lines (top) combined with human cell line mutation data (bottom). Red font indicates three cell lines identified carrying mutant NRAS.

B–E    Eleven different mouse and human tumor cell lines with known KRAS, NRAS, HRAS, EGFR, BRAF, PIK3CA, TRP53, and STK11 mutation status were injected s.c. in appropriate host mice (0.5 × 10^6 mouse and 10^6 human cells; n of cell lines is given in D and of mice in E). Primary tumor volume was monitored weekly and the animals were killed for macroscopic and microscopic lung examination when terminally ill. Shown are representative images of intravascular tumor emboli, micrometastases (red arrows) and macrometastases (black arrows) (B), representative lung stereoscopic images (C), summary of spontaneous lung metastatic behavior (D), and number (graph) and incidence (table) of macrometastases (E). Note visible B16F10 micrometastases expressing melanin (B).

Data information: Cell lines are described in the text. NRAS-mutant cells are in red font. Data are presented as mean ± SEM. P: probability by Fisher's exact test (D), one-way ANOVA (E, graph), or chi-square test (E, table). ***: P < 0.001 for comparison between any NRAS-wild-type and NRAS-mutant cell line by Bonferroni post-test (E, graph) or Fisher's exact test (E, table).

Source data are available online for this figure.

spontaneously colonizing the lungs from ectopic s.c. sites (Figs 3B and D, and EV2F). Even benign HEK293T cells delivered either s.c. or i.v. were rendered metastatic to the lungs of NOD/SCID mice (Blunt *et al*, 1995) upon *NRAS*^G61K transfection, as compared with p*C*-expressing cells that caused mere emboli and micrometastases (Figs 3B, C and E, and EV2G). In reverse experiments, lentiviral shRNA-mediated stable silencing of *Nras* (sh*Nras*) in *Nras*^Q61H-mutant LLC and AE17 cells was done. Since these also carried *Kras*^G12C mutations, side-by-side silencing of *Kras* (sh*Kras*) and of no known target (control shRNA; sh*C*) using otherwise

identical vectors was performed and the efficacy and specificity of this approach were validated (Figs 4A and B, and EV3). sh*Nras* exerted specific anti-metastatic effects, since it rendered s.c.-implanted LLC and AE17 cells virtually incapable for spontaneous lung metastasis without impacting primary tumor growth, while sh*Kras* reduced both primary tumor growth and pulmonary metastasis compared with sh*C* (Figs 4C and EV2H and I). Moreover, LLC and AE17 cells (*Nras*^Q61H) readily colonized the lungs upon intravenous (i.v.) delivery compared with MC38 cells (*Nras*^WT), a propensity that was also abrogated by sh*Nras*

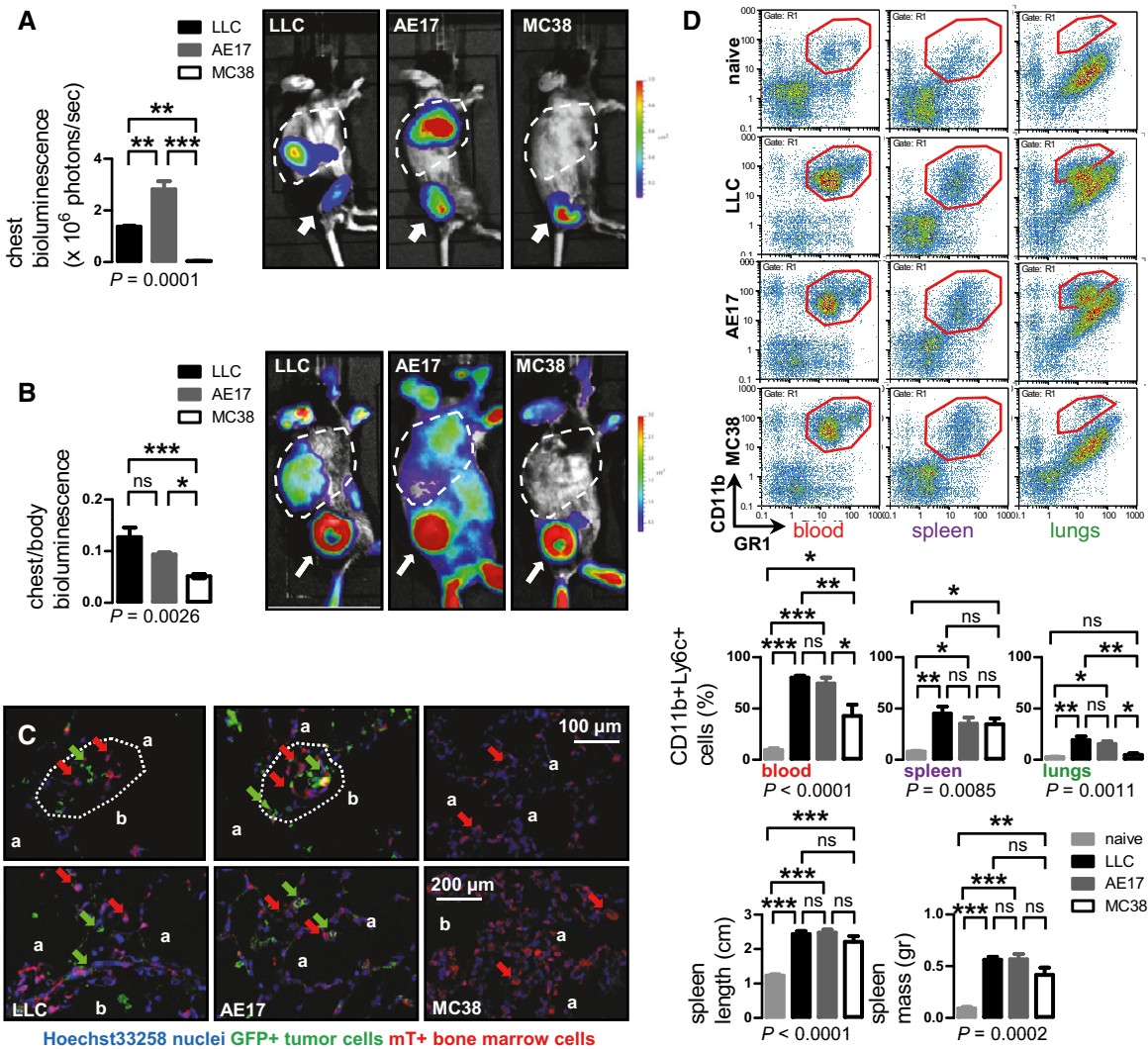

**Figure 2. Lung colonization by *NRAS*-mutant tumor cells is associated with local accumulation of myeloid cells.**

A, B  Bioluminescent images and data summaries of C57BL/6 mice with s.c. tumor cells expressing p*CAG.Luc* (A; *n* = 3/group) and of irradiated C57BL/6 chimeras reconstituted with *CAG.Luc.eGFP* bone marrow (B; *n* = 5) at 4 weeks after s.c. delivery of 10⁶ tumor cells. Arrows indicate primary tumors; primary tumors of experiment (A) were excised at 2 weeks post-injection. Dashed lines denote the thorax.

C  Lung sections of irradiated C57BL/6 mice, reconstituted with mT⁺ bone marrow and injected s.c. with 10⁶ tumor cells overexpressing p*eGFP* (*n* = 5/group) at 2 weeks post-tumor cells. Note the association of metastatic GFP⁺ tumor cells (green arrows) with mT⁺ bone marrow cells (red arrows) in niches (dashed outlines). Note also that mT⁺ myeloid cells in LLC- and AE17-colonized lungs were mononuclear and polymorphonuclear, while they were mononuclear in the lungs of mice with MC38 tumors. b, bronchus; a, alveolus.

D  Blood, spleen, and lung CD11b⁺Ly6c⁺ cells and splenic dimensions of C57BL/6 mice (*n* = 4/group) which received s.c. saline (naïve) or tumor cells. Similar results were obtained using CD11b⁺Gr1⁺ and CD11b⁺Ly6g⁺ (not shown).

Data information: Cell lines are described in the text. Data are presented as mean ± SEM. *P*: probability by one-way ANOVA. ns, *, **, and ***: *P* > 0.05, *P* < 0.05, *P* < 0.01, and *P* < 0.001, respectively, for indicated comparisons by Bonferroni post-test.

(Fig 4D). Collectively, these results show that *NRAS*^MUT and/or overexpression of *NRAS*^WT promotes lung colonization by circulating tumor cells in mouse models.

### *NRAS* drives chemokine signaling to pulmonary endothelial and myeloid chemokine receptors

To identify the downstream effectors of *NRAS*, RNA of sh*C*- and sh*Nras*-expressing LLC and AE17 cells (*Nras*^Q61H) was hybridized to

Affymetrix Mouse Gene ST2.0 microarrays (Fig EV4). Interestingly, the top sh*Nras*-suppressed transcripts were *Cxcl5* and *Ppbp* (encoding C-X-C-motif chemokine ligand 5, CXCL5, and pro-platelet basic protein, PPBP, respectively). Of note, these are the mouse orthologues of human interleukin-8 (IL-8) family members *CXCL6*, *CXCL7*, and *CXCL8* (Waugh & Wilson, 2008; Zlotnik & Yoshie, 2012), and the results were validated by qPCR (Fig 5A and B; Appendix Tables S1–S4). p*NRAS*^G61K-expressing MC38 and HEK293T cells also overexpressed *Cxcl5* and its human orthologue

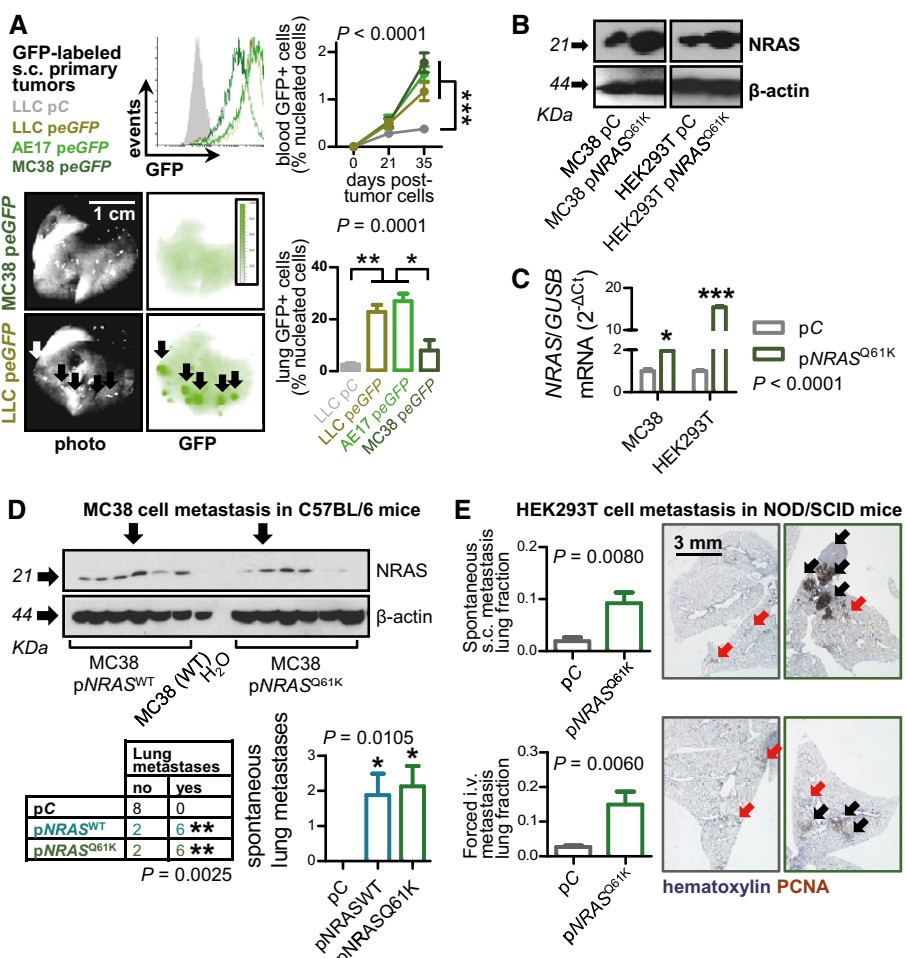

**Figure 3. Mutant *NRAS* promotes lung colonization by circulating tumor cells.**

A    Serial circulating and lung-homed tumor cells at 4 weeks after s.c. injection of $10^6$ peGFP-expressing tumor cells (*n* = 4/group). Histogram: flow cytometry of tumor cells. Images: representative lung photographs and biofluorescence at 445- to 490-nm excitation and 515- to 575-nm emission.

B, C    NRAS mRNA and protein of MC38 and HEK293T cells expressing control (pC), wild-type *NRAS* (p*NRAS*^WT), and *NRAS*^Q61K (p*NRAS*^Q61K) plasmids by qPCR (*n* = 3; shown are *NRAS* relative to *Gusb* or *GUSB* levels) and immunoblotting (shown is one representative of three experiments).

D    NRAS and β-actin immunoblots, incidence table, and multiplicity (graph) of lung metastases of C57BL/6 mice at 4 weeks after s.c. delivery of $10^6$ MC38 cells expressing pC, p*NRAS*^WT, and p*NRAS*^Q61K plasmids (*n* is given in table; arrows indicate the clones used for experiments).

E    Lung fraction occupied by metastases (graphs), and exemplary images of proliferating cell nuclear antigen (PCNA)-stained lung sections of NOD/SCID mice at 11 weeks after s.c. (*n* = 7–8/group) and i.v. (*n* = 5/group) delivery of $3 \times 10^6$ HEK293T cells expressing pC and p*NRAS*^Q61K.

Data information: Cell lines are described in the text. Arrows: micrometastases (red) and macrometastases (black). All data are presented as mean ± SEM. *P*: probability by two-way ANOVA (A, dot plot), one-way ANOVA (A, C, D, bar graphs), Fisher's exact test (D, table), or Student's *t*-test (E). *, **, and ***: *P* < 0.05, *P* < 0.01, and *P* < 0.001, respectively, for indicated comparisons (or for comparison with pC in C and D) by Bonferroni post-test.

Source data are available online for this figure.

*CXCL6* compared with pC-expressing counterparts, and SKMEL2 cells (*NRAS*^G61R) expressed higher levels of CXCL6 compared with A549 cells (*NRAS*^WT), indicating that mutant *NRAS* drives expression of IL-8-related chemokines (Fig 5C and D). We next examined the expression of *Cxcl5*, *Ppbp*, and their cognate receptor genes, *Cxcr1* and *Cxcr2* (Zlotnik & Yoshie, 2012) in tumor cells and metastasis target organs of our models. *Cxcl5* was expressed exclusively by tumor cells and *Ppbp* by tumor and host lung, myeloid, and liver cells. However, *Cxcr1* was expressed predominantly in the lungs and *Cxcr2* in the bone marrow (Fig 6A). Immunolocalization of CXCR1/2 proteins showed that CXCR1 was primarily expressed by endothelial cells of pulmonary blood vessels and CXCR2 by bone

marrow cells in lacunae (Fig 6B). Co-staining of lungs harboring spontaneous metastases for CXCR1 or CXCR2 and the endothelial marker CD34 identified native lung CXCR1^+CD34^+ endothelial cells and bone marrow-derived CXCR2^+CD34^+ endothelial cells, with the latter incorporating into metastasis-adjacent vessels and infiltrating lung tumors (Fig 6C). To pin the lineages that express CXCR1 and CXCR2 in lungs colonized by *NRAS*^MUT tumor cells, we intercrossed hematopoietic *Cre*-driver (*Vav.Cre*; Ogilvy *et al*, 1998) to *Cre*-reporter (*mT/mG*; Muzumdar *et al*, 2007) mice and pulsed offsprings with s.c. *NRAS*^MUT tumor cells to generate a lung metastasis model that features mG^+ hematopoietic, mT^+ non-hematopoietic, and non-fluorescent tumor cells. Indeed, CXCR1 was

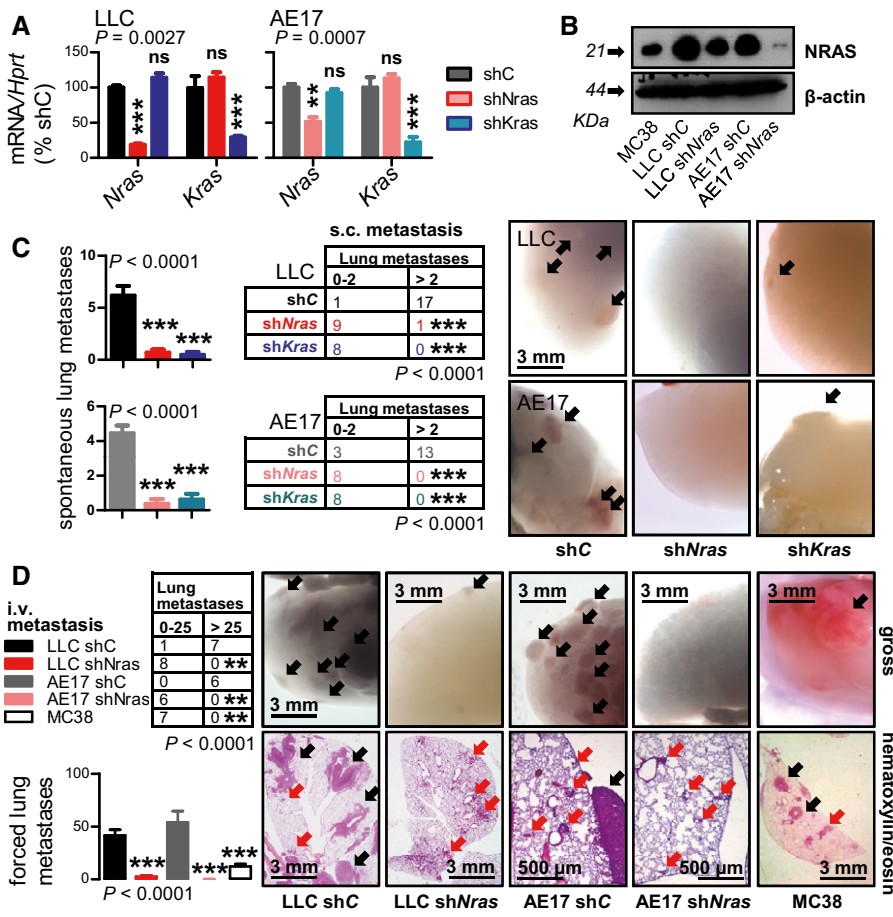

**Figure 4. Mutant *NRAS* is required for lung colonization.**

A, B    NRAS mRNA (A) and protein (B) expression of *Nras*[Q61H]-mutant mouse tumor cells (LLC and AE17) stably expressing control (sh*C*), anti-*Nras* (sh*Nras*)-, and anti-*Kras* (sh*Kras*)-specific shRNAs by qPCR (*n* = 3; shown are *Nras* or *Kras* relative to *Hprt* levels as percentage of control) and immunoblotting (shown is one representative of three experiments).

C    Incidence (tables), multiplicity (graphs), and exemplary images of automatic lung metastases (arrows) of C57BL/6 mice at 4 weeks after s.c. delivery of $10^6$ sh*C*-, sh*Nras*-, and sh*Kras*-expressing LLC and AE17 cells (*n* is given in tables).

D    Incidence (table), multiplicity (graph), and exemplary images of forced lung metastases of C57BL/6 mice at 2 weeks after i.v. delivery of $0.25 \times 10^6$ sh*C*- and sh*Nras*-expressing LLC and AE17 cells, as well as MC38 cells (*n* is given in table).

Data information: Cell lines are described in the text. Arrows: micrometastases (red) and macrometastases (black). All data are presented as mean ± SEM. *P*: probability by one-way ANOVA (graphs) or chi-square test (tables). ns, \*\*, and \*\*\*: *P* > 0.05, *P* < 0.01, and *P* < 0.001, respectively, for comparison with sh*C* by Bonferroni post-test (graphs) or Fisher's exact test (tables).

Source data are available online for this figure.

exclusively expressed by non-hematopoietic cells, whereas CXCR2 by hematopoietic cells (Fig 6D). To test the functional involvement of host CXCR1 and CXCR2 in pulmonary metastasis, single- and dual-copy *Cxcr1*-gene-deficient mice (Sakai *et al*, 2011), as well as single-copy *Cxcr2*-gene-deficient mice (homozygotes failed to thrive; Cacalano *et al*, 1994) received s.c. or i.v. LLC and AE17 cells (*Nras*[Q61H]). Both CXCR1 and CXCR2 were required for both spontaneous and induced lung colonization by these cells, but not for s.c. tumor growth (Figs 7A and B, and EV2J and K). In addition, BMT to lethally irradiated C57BL/6 or homozygote *Cxcr1*-deficient recipients (*Cxcr2*-haploinsufficient mice failed to recover after irradiation despite bone marrow transfer; *n* = 20) from C57BL/6, homozygote *Cxcr1*-deficient, or heterozygote *Cxcr2*-deficient donors identified that myeloid CXCR2, but not CXCR1, is required for lung

colonization by *Nras*[Q61H] LLC cells (Fig 7C). Taken together, these results indicate that *NRAS* mutations drive lung metastasis via IL-8-related chemokine interactions with lung endothelial CXCR1 and myeloid CXCR2 receptors.

## Implications of NRAS-driven chemokine signaling for human metastasis to the lungs

The potential translational value of our observations was probed, including the druggability of the proposed signaling axis and its potential existence in humans. To this end, combined inhibition of CXCR1 and CXCR2 signaling statistically and biologically significantly inhibited forced lung colonization by two different tumor cell lines with *Nras*[Q61H] alleles (Fig 8A). In addition, *NRAS*

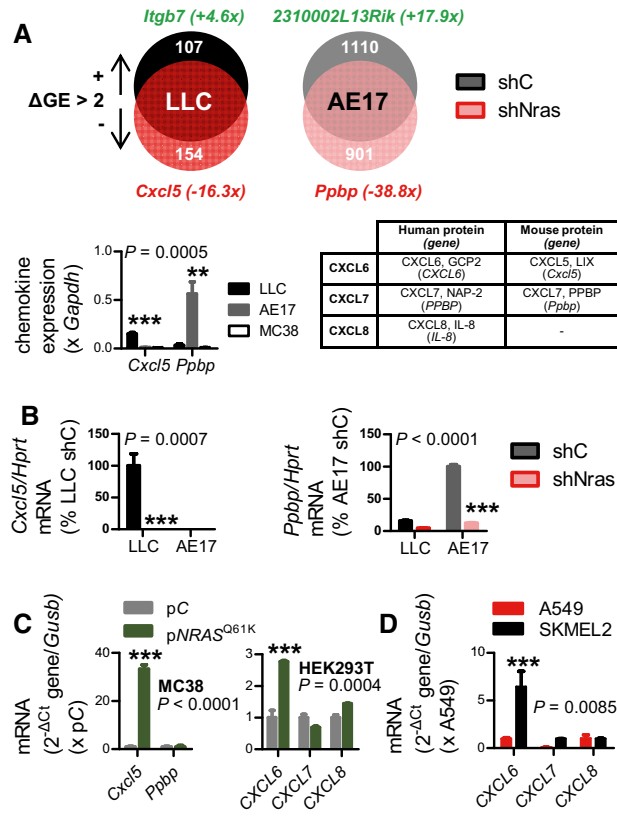

**Figure 5.  CXCR1/2 ligands as candidate effectors of mutant NRAS.**

A, B  Microarray (A) and qPCR (B) comparing differential gene expression (ΔGE) of Nras[Q61H]-mutant mouse tumor cells (LLC and AE17) expressing control (shC) or anti-Nras (shNras) shRNAs (n = 3). Table lists homologues of mouse and human CXCR1/2 ligand transcripts (Zlotnik & Yoshie, 2012).

C  CXCR1/2 ligand mRNA of MC38 and HEK293T cells stably expressing control (pC) and NRAS[Q61K] (pNRAS[Q61K]) plasmids by qPCR (n = 3) shows induction of Cxcl5 in MC38 cells and of CXCL6 in HEK293T cells.

D  CXCR1/2 ligand expression of A549 cells (NRAS wild type) and SKMEL2 cells (NRAS[Q61R]) by qPCR (n = 3) shows CXCL6 overexpression by SKMEL2 cells.

Data information: Cell lines are described in the text. All data are presented as mean ± SEM. P: probability by two-way ANOVA. ** and ***: P < 0.01 and P < 0.001, respectively, for comparison with all other groups (A), with shC (B), with pC (C), and with A549 cells (D) by Bonferroni post-test.

gain-of-function rates (from COSMIC; Forbes et al, 2015) were highly significantly correlated with the incidence of pulmonary metastases across bodily tumor from a large published necropsy study (Disibio & French, 2008) in a weighted analysis (Fig 8B). These data suggested that NRAS-driven lung metastasis can be indirectly targeted by inhibition of chemokine signaling and may exist in humans.

# Discussion

Here, we describe how NRAS gain of function promotes lung metastasis in mice by facilitating pulmonary homing of circulating tumor cells. We determine that both mutant NRAS and overexpressed wild-type NRAS are required and sufficient for spontaneous

pulmonary colonization by ectopic model tumors and their disseminated cells, irrespective of their tissues of origin and their KRAS mutation status. Moreover, we define that NRAS effects are delivered in a paracrine fashion via upregulation of tumor-secreted chemokines that signal to cognate CXCR1 receptors on lung endothelial and CXCR2 receptors on myeloid cells (Fig 8C). In addition to improvising mouse models of spontaneous lung metastasis on syngeneic backgrounds, we provide evidence for NRAS-mediated lung metastasis in mice and data suggesting this effect may occur in humans. Finally, we provide proof-of-concept results supporting targeting of NRAS-driven chemokine signaling as an effective means to limit lung colonization.

But can a single oncogene cause lung metastasis? Are clinical observational data concordant to this stipulation in general (Tie et al, 2011; Urosevic et al, 2014; Lan et al, 2015; Pereira et al, 2015; Ulivieri et al, 2015)? A link between NRAS gain and lung metastasis is plausible, since both occur across human cancer types (Forbes et al, 2015), NRAS mutations/gain occurs prior to the onset of metastasis in melanoma (Edlundh-Rose et al, 2006; Colombino et al, 2012), and since NRAS mutations/gain is especially frequent in cancers that commonly home to the lungs (i.e. melanoma, thyroid and colon adenocarcinomas, acute myeloid leukemia, and metastatic tumors of unknown primary). In addition, oncogenic NRAS has been mainly studied in respect to cell-autonomous processes (Whitwam et al, 2007; Mandala et al, 2014), and tumor–host interactions triggered by the oncogene may have gone under-appreciated. We provide evidence from mice for a causative association of NRAS mutations/gain and metastatic pulmonary tropism. First, different murine and human tumor cells sharing codon 61 NRAS mutations presented uniform pulmonary tropism independent from coexisting mutations and tissue of origin. Second, mutant NRAS was required for spontaneous homing to the lungs of two cancer cell lines and NRAS mutation and/or overexpression was sufficient to transmit this phenotype to NRAS-wild-type tumor cells, as well as to benign cells. Third, a close correlation between NRAS mutation/gain rates and lung metastasis was identified in large published human datasets. We also show that NRAS mutations are associated with overexpression of the oncogene in accord with human cancer data (Stephen et al, 2014; Pfarr et al, 2016) and that overexpression of NRAS[WT] also suffices for lung metastasis.

In addition to identifying a connection between NRAS and lung metastasis, our data provide mechanistic clues on the stage of the metastatic process at which NRAS mutation/gain becomes pivotal (Valastyan & Weinberg, 2011; Vanharanta & Massague, 2013): The oncogene appears to be dispensable for tumor cell dissemination from the primary tumor, but to be cardinal for homing of circulating tumor cells to the lungs. Indeed, labeled primary tumor cells disseminated into the blood independent from NRAS mutation status, but the latter was pivotal for pulmonary involvement by both s.c. and i.v. injected tumor cells. The proposed late-stage pro-metastatic functions of NRAS are compatible with its proposed impact across bodily tumor types and adds potential clinical value to the findings, implying it can be a marked therapeutic target against lung metastases from any tumor, regardless of the divergent mechanisms employed by different cancers to gain access to the vasculature (Rafii et al, 2002; Valastyan & Weinberg, 2011; Vanharanta & Massague, 2013).

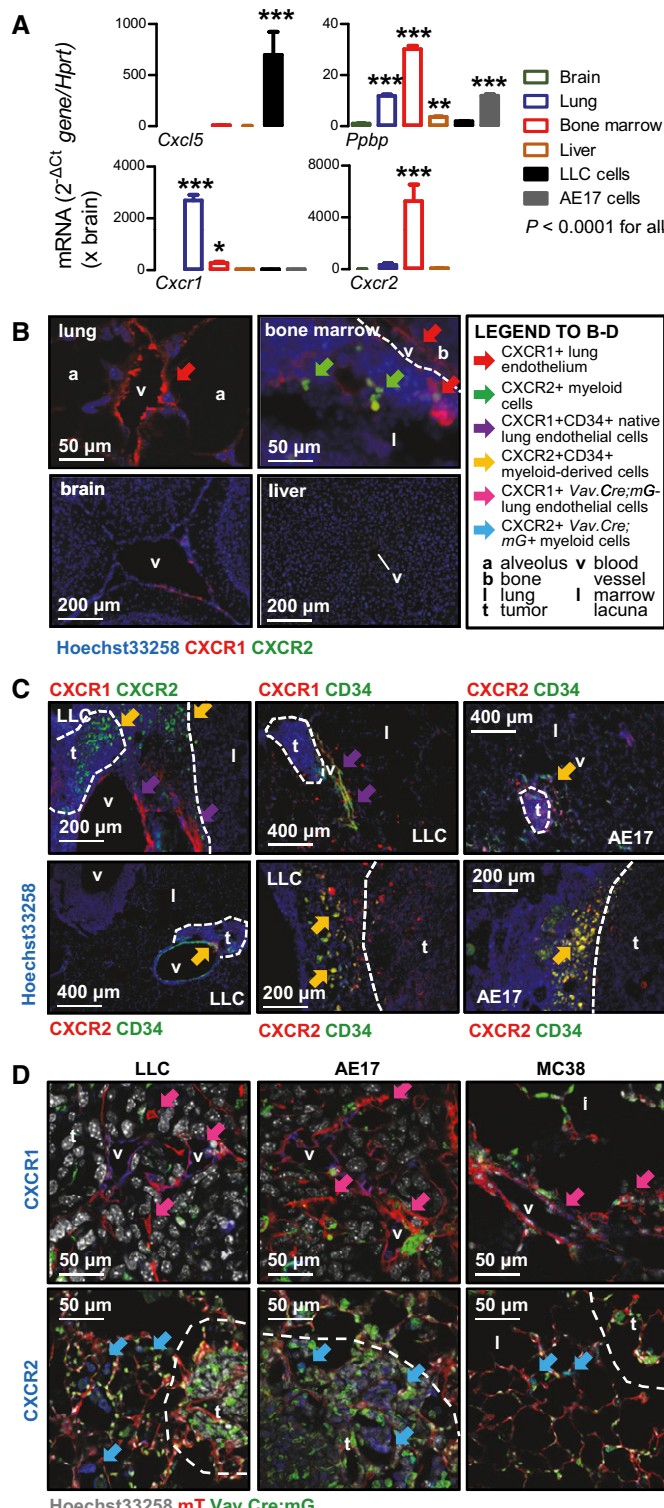

**Figure 6.  High-level CXCR1 expression of the pulmonary endothelium and CXCR2 expression of bone marrow cells.**

A   *Cxcr1*, *Cxcr2*, *Cxcl5*, and *Ppbp* mRNA expression in metastatic target organs and *Nras*[Q61H]-mutant tumor cells (LLC and AE17) of C57BL/6 mice by qPCR (*n* = 3).

B   Immunostaining of tumor-free mouse organs for CXCR1 and CXCR2. Lung, bone marrow, brain and hepatic vessels, but not lung alveoli, expressed CXCR1, whereas CXCR2 was expressed by bone marrow cells. Dotted lines indicate interface between bone and bone marrow lacuna.

C   Immune co-labeling of mouse lungs with spontaneous metastases from s.c. *Nras*[Q61H]-mutant tumor cells (LLC, AE17) for CXCR1, CXCR2, and the endothelial marker CD34 identified both native lung CXCR1[+]CD34[+] bone marrow-derived CXCR2[+]CD34[+] endothelial cells, with the latter incorporating into metastasis-adjacent vessels and infiltrating lung tumors.

D   Immunofluorescent staining for CXCR1 and CXCR2 of lungs from *Vav.Cre; mT/mG* mice pulsed s.c. with LLC, AE17, and MC38 tumor cells revealed CXCR1 expression in mT[+] cells and CXCR2 expression in GFP[+] cells.

Data information: Cell lines are described in the text. Data are presented as mean ± SEM. *P*: probability by one-way ANOVA. *, **, and ***: *P* < 0.05, *P* < 0.01, and *P* < 0.001, respectively, for comparison with brain by Bonferroni post-test. (C, D): Dotted lines indicate tumor boundaries.

tumor growth. Although the functional association of *NRAS* mutations/gain with metastatic propensity merits further investigation and is likely linked to both cell-autonomous and paracrine signaling of cancer cells (Valastyan & Weinberg, 2011; Vanharanta & Massague, 2013), our results support that pulmonary tropism induced by mutant *NRAS* is predominantly mediated via tumor–host interactions *in vivo*.

The notion that chemokine secretion by circulating tumor cells can destine their fate is also pioneering. We show that *NRAS*-mutated cancer cells exhibit predilection for the lungs via expression of chemokine ligands and not receptors. It has been established that disseminated tumor cells display organ tropism by expressing cell surface molecules that facilitate adhesive interactions, rather than solute inflammatory and vasoactive mediators (Abdel-Ghany *et al*, 2001; Brown & Ruoslahti, 2004; Auguste *et al*, 2007). However, chemokine gradients could conceivably facilitate adhesion of tumor cells lodged in the tight pulmonary arterioles and capillaries to the CXCR1-expressing pulmonary endothelium. In addition, chemokine ligands may activate the pulmonary endothelium during this transit, facilitating their trespassing into the interstitium (Schraufstatter *et al*, 2001; Wolf *et al*, 2012). Conceivably, elevated systemic levels of CXCR1 ligands secreted by circulating tumor cells may precondition the lung endothelium, thereby enhancing its receptivity to tumor cells. CXCR1/2 and their ligands have been implicated in angiogenesis, tumor dissemination, myeloid cell recruitment, and self-seeding of primary tumors, but their potential impact in metastatic organ tropism is less well defined (Waugh & Wilson, 2008). Similar to other known paracrine molecular targets of RAS that have been shown to foster extravasation (Rak *et al*, 2000; Gupta *et al*, 2007; Qian *et al*, 2011), we show that *NRAS* impacts tumor cell homing to the lungs likely via CXCR1/2 signaling. Although we do not directly show that chemokines are responsible for *NRAS*-mediated pulmonary metastasis, this is highly likely based on the data. Interestingly, CXCR1 expression was largely restricted to the lung endothelium, and *NRAS*-mutant cancer cells exhibited a specific pulmonary metastatic pattern likely dictated by their chemokine expression profile, similar to other pro-metastatic axes (Padua *et al*, 2008; Zhang *et al*, 2009).

Another interesting finding is the unanticipated paracrine mode of *NRAS* effects during pulmonary metastasis. To this end, silencing of *NRAS* in two tumor cell lines of different origins abolished the expression of two closely related chemokines. Moreover, silencing and overexpression of *NRAS* across different cancer cell lines selectively impacted metastatic capacity to the lungs, but not primary

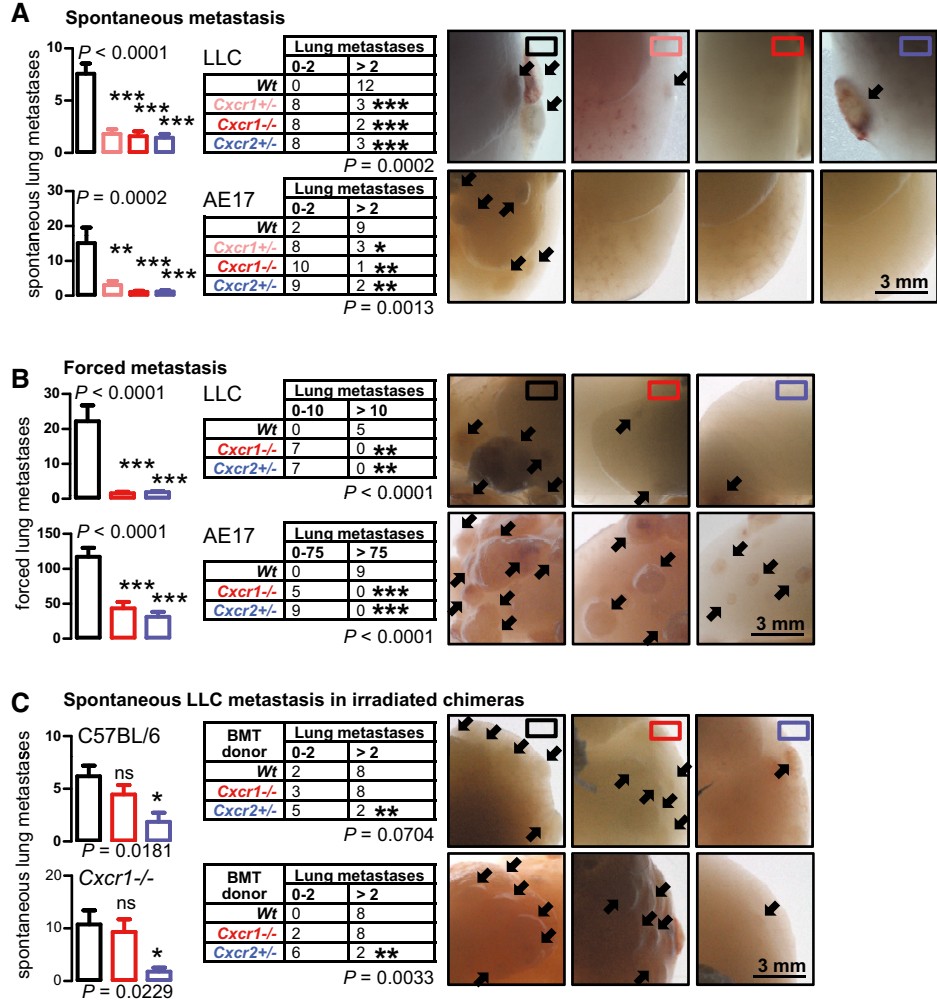

**Figure 7. Non-myeloid CXCR1 and myeloid CXCR2 are required for spontaneous lung metastasis.**

A, B    Incidence (tables), multiplicity (graphs), and exemplary images of lung metastases (arrows) of LLC and AE17 cells in C57BL/6, single- and dual-copy *Cxcr1*-gene-deficient (*Cxcr1^+/−^* and *Cxcr1^−/−^*, respectively), and single-copy *Cxcr2*-gene-deficient (*Cxcr2^+/−^*) mice at 4 weeks after s.c. delivery of 10^6 cells (A) and at 2 weeks after i.v. delivery of 0.25 × 10^6 cells (B). *n* is given in tables.

C    Incidence (tables), multiplicity (bar graphs), and exemplary images of automatic lung metastases (arrows) of irradiated C57BL/6 and *Cxcr1^−/−^* chimeras reconstituted with bone marrow from C57BL/6, *Cxcr1^−/−^*, or *Cxcr2^+/−^* donors at 4 weeks after s.c. delivery of 10^6 LLC cells.

Data information: Cell lines are described in the text. *n* is given in tables. All data are presented as mean ± SEM. *P*: probability by one-way ANOVA (graphs) or chi-square test (tables). ns, *, **, and ***: *P* > 0.05, *P* < 0.05, *P* < 0.01, and *P* < 0.001, respectively, for comparison with wild-type mice (A, B) or donors (C) by Bonferroni post-test (graphs) or Fisher's exact test (tables).

Source data are available online for this figure.

Although they cannot shed light on the sequence of events that lead to the formation of the lung premetastatic niche, our results emphasize the importance of myeloid cells in this process. Using BMT, we show that CXCR1-mediated tumor cell–endothelial interactions are not sufficient for *NRAS*-driven lung metastasis, which also requires the concurrent involvement of myeloid cells that express CXCR2. We show that these cells infiltrate and possibly predispose the local microenvironment for the successful settling of disseminated *NRAS*-mutant tumor cells, and include CD45^+CD11b^+Ly6c^+ and CD45^+CD11b^+Ly6g^+ hematopoietic as well as CXCR2^+CD34^+ endothelial progenitors. *NRAS*-driven chemokine signaling by lung-extravasated tumor cells is likely responsible for the homing of myeloid cells to the lungs in our models. Whichever comes first,

niche formation by carcinoma and myeloid cells appears to be critical for the remodeling of the local tissues and for sustained colonization of the target organ (Waugh & Wilson, 2008; Psaila & Lyden, 2009).

CXCR1 and CXCR2 signaling has been associated with the progression of various primary tumors and has been the focus of several therapeutic efforts (Sun *et al*, 2003; Sharma *et al*, 2010; Khan *et al*, 2015). Our evidence suggests that these receptors also directly impact metastasis, as we found that they are required to fine-tune the sustained lodging and outgrowth of *NRAS*-mutant circulating tumor cells in the lungs. More importantly, we show that CXCR1/2-mediated pneumotropism can be annihilated, since navarixin significantly prevented lung tumor colonies sprouted by circulating *NRAS*-mutant tumor cells. Navarixin has been successful in alleviating

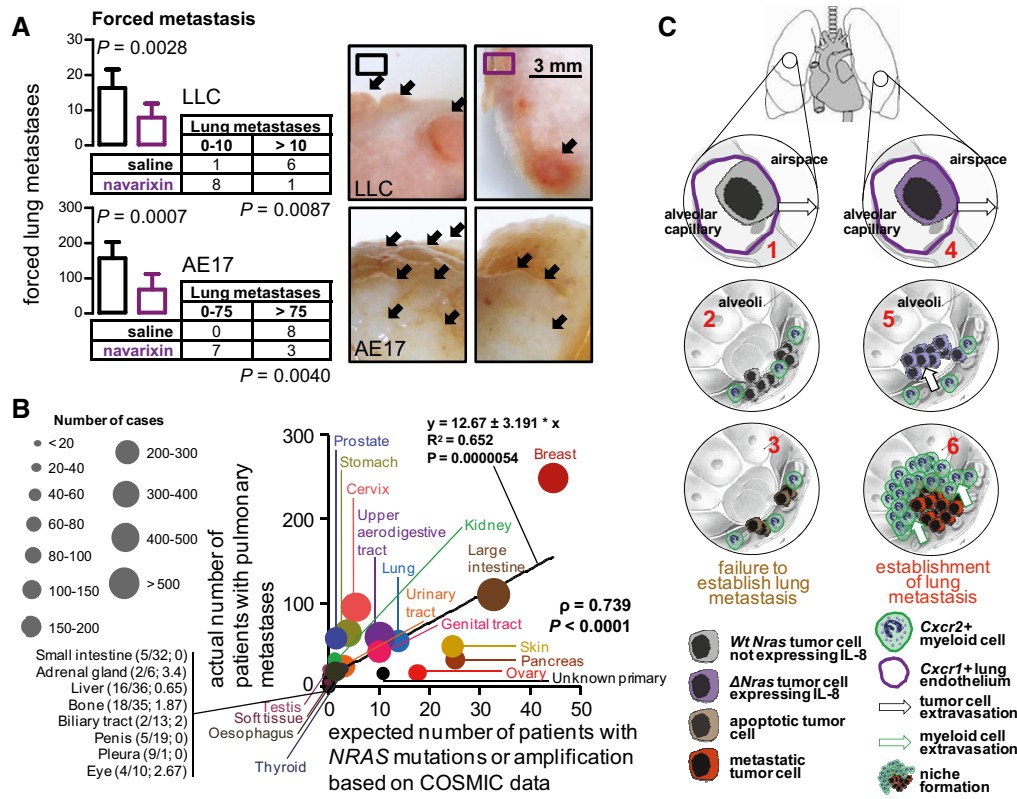

**Figure 8.  Targeting CXCR1/2 prevents pulmonary metastasis by circulating *NRAS*-mutant tumor cells and NRAS perturbations correlate with lung metastasis in humans.**

A   Incidence (tables), multiplicity (graphs), and exemplary images of lung metastases (arrows) of C57BL/6 mice at 2 weeks after i.v. delivery of $0.25 \times 10^6$ LLC or AE17 cells preceded by 5 h and followed by 7 days by oral gavage with 200 μl saline or the CXCR1/2 antagonist navarixin (300 μg in 200 μl saline). Data are presented as mean ± SEM. *P*: probability by Fisher's exact (tables) or Student's *t*-test (graphs). *n* is given in tables.

B   Number of patients with lung metastases (*n* = 1,015) identified at necropsy among 3,817 patients with 26 different bodily tumor types (from Disibio & French, 2008) plotted versus the number of patients expected to harbor *NRAS* gain of function (*n* = 216; calculated from COSMIC; Forbes *et al*, 2015). Shown are weighted data points (bubble size indicates total number of patients with each tumor type); Spearman's correlation coefficient (ρ) and probability value (*P*); and linear regression line with formula, Pearson's coefficient (R), and probability value (*P*).

C   Mutant *NRAS* functions in lung metastasis. *NRAS*-mutant circulating tumor cells express CXCR1 ligands and attach to the CXCR1[+] pulmonary endothelium (4). Lung-homed *NRAS*-mutant tumor cells systemically shed CXCR1 ligands to recruit CXCR2[+] myeloid cells to form pulmonary niches (5) and to foster metastasis (6). Wild-type *NRAS* tumor cells do not express CXCR1 ligands and fail to attach, to recruit myeloid cells, and establish lung metastases (1–3).

Source data are available online for this figure.

pulmonary inflammation in patients with chronic obstructive pulmonary disease (Rennard *et al*, 2015) and would be worth testing in patients at risk of lung metastasis from *NRAS*-mutant tumors.

In summary, we report here for the first time that *NRAS* mutations and/or gain of function drive lung metastasis in mouse models and that this metastatic trait can be indirectly blocked by chemokine receptor antagonism. We believe that these findings are worth exploring in the setting of clinical trials designed to prevent or cure lung metastases in patients with *NRAS*-mutant primary tumors.

# Materials and Methods

## Cell culture

Cells were cultured at 37°C in 5% $CO_2$-95% air using DMEM supplemented with 10% FBS, 2 mM L-glutamine, 1 mM pyruvate,

100 U/ml penicillin, and 100 mg/ml streptomycin. Cell lines were tested annually for identity (by the short tandem repeat method) and for *Mycoplasma* spp. (by PCR). For *in vivo* injections, cells were harvested using trypsin, incubated with Trypan blue, and counted, as described elsewhere (Marazioti *et al*, 2013; Giannou *et al*, 2015). Only 95% viable cell populations were used for *in vivo* injections.

## Cancer cell lines

C57BL/6 mouse Lewis lung carcinoma (LLC), B16F10 skin melanoma, and PANO2 pancreatic adenocarcinoma, BALB/c mouse C26 colon adenocarcinoma; and human A549 and HCC-827 lung adenocarcinomas, SKMEL2 skin melanoma, and MDA-MB-231 breast adenocarcinomas were from the National Cancer Institute's Tumor Repository (Frederick, MD). Human HEK293T embryonic kidney cells were from the American Type Culture Collection (Manassas, VA). C57BL/6 mouse MC38 colon adenocarcinoma cells were a gift

from Dr. Barbara Fingleton (Vanderbilt University, Nashville, TN; Marazioti *et al*, 2013; Giannou *et al*, 2015) and C57BL/6 mouse AE17 and BALB/c mouse AB2 malignant pleural mesothelioma cells a gift from Dr. YC Gary Lee (University of Western Australia, Perth, Australia; Jackaman *et al*, 2003).

## Sanger sequencing

Total cellular RNA was isolated using TRIzol (Thermo Fisher Scientific, Waltham, MA) followed by RNeasy purification and genomic DNA removal (Qiagen, Hilden, Germany). One μg total RNA was reverse-transcribed using Oligo(dT)$_{18}$ primers and Superscript III (Thermo Fisher Scientific) according to manufacturer's instructions. For sequencing reactions, *Kras, Nras, Hras, Egfr, Braf, Pik3ca, Trp53,* and *Stk11* cDNAs were amplified in PCRs using primers of Appendix Table S5 and Phusion Hot Start Flex polymerase (New England Biolabs, Ipswich, MA). DNA fragments were purified with NucleoSpin gel and PCR cleanup columns (Macherey-Nagel, Düren, Germany) and were sequenced using their primer sets by VBC Biotech (Vienna, Austria).

## Quantitative real-time PCR (qPCR)

Cellular RNA was retrieved as above. For tissue RNA extracts, brain, lung, and liver (but not bone marrow) tissues were passed through 70-μm strainers (BD Biosciences, San Jose, CA), single cell suspensions were counted with a hemocytometer, and $10^7$ cells were subjected to RNA extraction as above. qPCR was performed using first-strand synthesis with specific primers (Appendix Table S5) and SYBR FAST qPCR Kit (Kapa Biosystems, Wilmington, MA) in a StepOne cycler (Applied Biosystems, Carlsbad, CA). $C_t$ values from triplicate reactions were analyzed with the relative quantification method $2^{-\Delta C_t}$ (Pfaffl, 2001). The abundance of a given mRNA was determined relative to glucuronidase beta (*Gusb, GUSB*) or hypoxanthine guanine phosphoribosyl transferase (*Hprt, HPRT*) reference transcripts. mRNA levels are presented as either $2^{-\Delta C_t} = 2^{-(C_t \text{ of transcript of interest})-(C_t \text{ of control transcript})}$ or as percentages or fractions relative to control samples. All qPCR analyses of cell lines were done on triplicate samples (technical $n = 3$) obtained on at least three independent occasions from each cell line (biological $n \geq 3$). All qPCR analyses of tissues were done on triplicate samples (technical $n = 3$) obtained from three different C57BL/6 mice (biological $n = 3$).

## Microarray

For microarray, triplicate cultures of $10^6$ cells (for each cell line/condition) were subjected to RNA extraction as above. Five μg of pooled total RNA was tested for quality on an ABI 2000 bioanalyzer (Agilent Technologies, Sta. Clara, CA), labeled, and hybridized to GeneChip Mouse Gene 2.0 ST arrays (Affymetrix, Sta. Clara, CA). Analyses was performed on the Affymetrix Expression Console (parameters: annotation confidence, full; summarization method: iter-PLIER include DABG; background: PM-GCBG; normalization method: none), followed by normalization using a LOWESS multi-array algorithm. Intensity-dependent estimation of noise was used for statistical analysis of differential expression. All data were deposited at GEO (http://www.ncbi.nlm.nih.gov/geo/; Accession ID: GSE74309).

## Mouse models of lung metastasis

C57BL/6 (#000664), BALB/c (#001026), NOD/SCID (#001303), *CAG.Luc.eGFP* (#008450), *mT/mG* (#007676), *Vav.Cre* (#008610), *Cxcr1*$^{-/-}$ (#005820), and *Cxcr2*$^{+/-}$ (#006848) mice were from Jackson Laboratories (Bar Harbor, MN) and were bred at the Center for Animal Models of Disease of the University of Patras. Experiments were designed and approved *a priori* by the Veterinary Administration of the Prefecture of Western Greece (approval numbers 3741/16.11.2010, 60291/3035/19.03.2012, and 118018/578/30.04.2014) and were conducted according to Directive 2010/63/EU (http://eur-lex.europa.eu/LexUriServ/LexUriServ.do?uri = OJ:L:2010:276:0033:0079:EN:PDF). The 611 male and female experimental mice used for these studies were sex-, weight (20–25 g)-, and age (6–12 weeks)-matched. For solid tumor induction, mice were anesthetized using isoflurane and received s.c. injections of 100 μl phosphate-buffered saline (PBS) containing $0.5 \times 10^6$ mouse cancer cells, $10^6$ human cancer cells, or $3 \times 10^6$ HEK293T cells. Three vertical tumor dimensions ($\delta^1$, $\delta^2$, $\delta^3$) were monitored longitudinally and tumor volume was calculated as $\delta^1\delta^2\delta^3/2$ (Marazioti *et al*, 2013; Giannou *et al*, 2015). Mice were sacrificed only when moribund, and specific permits were obtained to let tumors grow beyond 1.5 cm$^3$ since spontaneous lung metastases rarely occurred under this limit. For induction of forced lung metastases, mice received i.v. injections of 100 μl PBS containing $0.25 \times 10^6$ murine or $3 \times 10^6$ HEK293T cells. Mice were sacrificed after 2 or 11 weeks, respectively, as described elsewhere (Stathopoulos *et al*, 2008). Oral gavage of 300 μg of the dual CXCR1/2 receptor antagonist navarixin (SCH527123; ApexBio, Houston, TX; Dwyer *et al*, 2006) in 200 μl saline or 200 μl saline were given at 5 h pre- and 7 days post-tumor cell injection.

## Quantification of lung metastases

Lungs were fixed in 10% neutral buffered formalin overnight and the number of lung metastases was macroscopically evaluated by two independent and blinded investigators on a Stemi DV4 stereoscope (Zeiss, Jena, Germany). Numbers were averaged and were reevaluated when divergent by > 20%. Lungs were embedded in paraffin, randomly sampled by cutting 4-μm-thick lung sections ($n = 10$/lung), mounted on glass slides, and stained with hematoxylin and eosin. Lung tumor fraction, that is, the fraction of the lung volume occupied by metastases, was determined by point counting of the ratio of the area of metastases versus the lung area, as described previously (Hsia *et al*, 2010; Giopanou *et al*, 2016).

## Constructs

p*CAG.Luc* plasmid encoding constitutive *Photinus pyralis* luciferase (*Luc*) was constructed as follows and was deposited with Addgene (#74409). The vector pPY.CAG.CRE.IP (Tashiro *et al*, 1999) was digested with XbaI/BamHI and the 4,231-bp backbone was ligated to a 1,983-bp fragment containing *Luc* coding sequences taken from the BamHI/NheI-digested pGL3 basic reporter vector (Promega, Madison, WI). The resulting plasmid was linearized with BamHI, followed by ligation of a BglII/BamHI linear PCR product containing the puromycin resistance coding sequences under the phosphoglycerate kinase mammalian promoter (PGK). Control Babe-puro (Morgenstern & Land, 1990; Addgene #1764), wild-type

*NRAS*-encoding (pCGN N-Ras wt; Fiordalisi *et al*, 2001; Addgene #14723), or mutant *NRAS*$^{Q61K}$-encoding (Babe N-Ras 61K; Khosravi-Far *et al*, 1996; Addgene #12543) retroviral plasmids were from Addgene. Lentiviral pools of p*eGFP* (sc-108084), random (shC, sc-108080), anti-*Nras* (sh*Nras*, sc-36005-V), and anti-*Kras* (sh*Kras*, sc-43876-V) shRNAs were from Santa Cruz Biotechnology (Santa Cruz, CA). Target sequences of sh*Nras* were caaggacagttgacacaaa, ggaatagatgtgtcaagaa, and cccatcaccttgaaactaa and of sh*Kras* ctacaggaaacaagtagta, gaacagtagacacgaaaca, and ccattcagtttc catgtta (Fig EV3). For transfections, 20–30% confluent tumor cells cultured in six-well culture vessels were incubated with 1 μg plasmid DNA using Xfect (Clontech, Mountain View, CA) and clones were selected by puromycin (1–10 μg/ml). To package plasmids into retroviral particles, 50% confluent HEK293T cells cultured in 3-mm culture vessels were transfected with 5 μg plasmid DNA of the desired vector together with VSV-G envelope-expressing plasmid pMD2.G (gift from Didier Trono, Addgene #12259) and pCMV-Gag-Pol vector expressing the retroviral structure proteins (Cell Biolabs Inc, San Diego, CA) at 1.5:1:1 DNA stoichiometry via the CaCl$_2$/BES method. After 2 days, the culture medium (2 ml) was collected, passed through a 45-μm filter, supplemented with 8 ml fresh DMEM, and subsequently overlaid on a 100-mm vessel containing 30% confluent mouse cancer cells. After 48 h, the medium was removed and stable clones were isolated using puromycin selection as above.

### Imaging

Mice were imaged for bioluminescence after i.v. delivery of 1 mg D-luciferin (Gold Biotechnology, St. Louis, MO) and saline-inflated lungs for biofluorescence with the 445- to 490-nm excitation and 515- to 575-nm emission filters of a Xenogen Lumina II, and data were analyzed on Living Image v.4.2 (PerkinElmer, Waltham, MA; Giannou *et al*, 2015; Marazioti *et al*, 2013).

### Bone marrow transfer (BMT)

For adoptive BMT, C57BL/6 mice received $10^7$ bone marrow cells i.v. 12 h after total body irradiation (1,100 Rad). Full bone marrow reconstitution was completed after 1 month (Marazioti *et al*, 2013; Giannou *et al*, 2015).

### Histology, immunohistochemistry, and immunofluorescence

Lungs were inflated and fixed with 10% formalin overnight, embedded in paraffin, and stored at room temperature. Five-micrometer paraffin sections were mounted on glass slides and counterstained with hematoxylin and eosin (Sigma-Aldrich, St. Louis, MO) or processed for immunohistochemistry as follows: Sections were deparaffinized by ethanol gradient, rehydrated, and boiled in heat-induced epitope antigen retrieval solution (0.1 M sodium citrate; pH = 6.0). Endogenous peroxidase activity was quenched with 3% H$_2$O$_2$, and non-specific binding was blocked using Tris-buffered saline with 3% bovine-serum albumin. Sections were incubated with antibodies of Appendix Table S6 overnight at 4°C, and antibody detection was performed with Envision (Dako, Carpinteria, CA). Sections were counterstained with hematoxylin and mounted with Entellan new (Merck Millipore, Darmstadt, Germany). For immunofluorescence, lungs were inflated with a 2:1

mixture of 4% paraformaldehyde in optimal cutting temperature (OCT; Sakura, Tokyo, Japan), washed with PBS, fixed in 4% paraformaldehyde overnight at 4°C, washed again with PBS, cryoprotected with 30% sucrose overnight at 4°C, embedded in OCT, and stored at −80°C. Ten-micrometer cryosections were labeled with primary antibodies overnight at 4°C followed by incubation with fluorescent secondary antibodies (Appendix Table S6). Sections were counterstained with Hoechst 33258 (Sigma-Aldrich) and mounted with Mowiol 4–88 (Calbiochem, Gibbstown, NJ). For isotype control, primary antibody was omitted. Bright-field images were captured with an AxioLab.A1 microscope connected to an AxioCam ERc 5s camera (Zeiss). Fluorescent microscopy was carried out on an AxioObserver.D1 inverted (Zeiss) or a TCS SP5 confocal microscope (Leica Microsystems, Heidelberg, Germany) and digital images were processed with Fiji academic software (Schindelin *et al*, 2012).

### Flow cytometry

Mouse tissues (lungs, spleens, blood) were processed as follows: The lungs were isolated in cold DMEM, minced with a scalpel, passed through 70-μm cell strainers (BD Biosciences), and resuspended in FACS buffer (PBS supplemented with 2% FBS and 0,1% NaN$^3$; $10^6$ cells/50 μl). Spleens were underwent NH$_4$Cl red blood cells lysis before resuspension in FACS buffer. Blood was drawn with heparinized 26G syringes, processed for red blood cell lysis, and resuspended in FACS buffer. Cultured cell lines were washed with PBS, trypsinized, and suspended in FACS buffer. All cells were stained with the antibodies of Appendix Table S6 for 20 min in the dark and analyzed on a CyFlow ML with FloMax Software (Partec, GmbH, Munster, Germany).

### Immunoblotting

Total protein extracts from cultured cells were prepared using Mg$^{2+}$ lysis/wash buffer [25 mM HEPES (pH = 7.5), 150 mM NaCl, 1% NP-40, 10 mM MgCl$_2$, 1 mM EDTA, 2% glycerol]. Proteins were separated by 15% sodium dodecyl sulfate–polyacrylamide gel electrophoresis (SDS–PAGE) and were electroblotted to PVDF membranes (Merck Millipore). Membranes were probed with primary antibodies followed by incubation with appropriate horseradish peroxidase-conjugated secondary antibodies (Appendix Table S6) and were visualized with enhanced chemiluminescence (Merck Millipore).

### Statistics

A target sample size of $n = 10$/group was calculated using G*power (Faul *et al*, 2007) assuming α-error = 0.05, β-error = 0.95, and Cohen's $d = 1.5$ tailored to detect at least 30% differences between groups featuring standard deviations (SD) spanning 30% of the mean. No data were excluded from analyses. Animals were allocated to treatments by alternation and transgenic animals were enrolled case–control-wise. Data acquisition was blinded on samples previously coded by a non-blinded investigator. All data were examined for normality by Kolmogorov–Smirnov test and were distributed normally. Values are given as mean ± standard error of the mean (SEM) and variance between groups was similar. Sample size ($n$) refers to biological replicates. Differences in means were examined by *t*-test or one-way analysis of variance (ANOVA)

### The paper explained

#### Problem
Although *NRAS* mutations render tumors more prone to systemic dissemination, their role in lung metastasis has not been functionally addressed.

#### Results
We used an array of mouse model systems to identify that *NRAS* mutations or gain predisposes cancer cells to spontaneously colonizing the lungs. This link was functionally validated using direct modulation of NRAS expression. Interleukin-8-related chemokines were identified as the tumor cell-secreted culprits for NRAS-dependent pulmonary metastatic propensity, signaling to lung endothelial and myeloid cells to facilitate pulmonary invasion.

#### Impact
These findings bear implications for the follow-up of patients with *NRAS*-mutant primary tumors, for the prevention of pulmonary metastasis in such patients, and for future trial design in patients with solid tumors with high *NRAS* mutation frequencies, such as melanoma.

with Bonferroni post-tests and in frequencies by Fisher's exact or chi-square tests. Flank tumor growth was analyzed by two-way ANOVA with Bonferroni post-tests Correlations were done using Pearson's $R$ or Spearman's $\rho$. $P$-values are two-tailed, and $P < 0.05$ was considered significant. Analyses and plots were done on Prism v5.0 (GraphPad, La Jolla, CA).

### Accession numbers

Microarray data were deposited at GEO (http://www.ncbi.nlm.nih.gov/geo/; Accession ID: GSE74309) and p*CAG.Luc* plasmid with Addgene (#74409).

### Data availability

Primary metastasis data of human cancers were decoded from Fig 1 of Disibio and French (2008).

NRAS gain-of-function frequency data in the various human cancer types were retrieved from the Catalogue of Somatic Mutations in Cancer (http://cancer.sanger.ac.uk/cosmic/gene/analysis? ln = NRAS#dist; Forbes *et al*, 2015).

**Expanded View** for this article is available online.

### Acknowledgements

The authors thank the University of Patras Center for Animal Models of Disease and Advanced Light Microscopy Core for experimental support. This work was supported by European Research Council 2010 Starting Independent Investigator and 2015 Proof of Concept Grants (260524 and 679345, respectively, to GTS) and by a RESPIRE2 European Respiratory Society Fellowship (2015–7160 to IP).

### Author contributions

ADG, AM, and NIK conceived, designed, and carried out most experiments, analyzed the data, provided critical intellectual input, and wrote portions of the paper draft; IL and DEZ did tissue and cellular immune labeling, brightfield and confocal microscopy, and quantification; IG, GN, GAG, DaK, VA, ACK, and ML carried out mouse models of i.v. lung metastasis, expression profiling of cell lines, and mutant *NRAS* overexpression studies in wild-type and transgenic mice; TA performed mutant *NRAS* silencing/overexpression and relevant *in vitro* and *in vivo* assays; MV and MP performed transgenic animal intercrosses and experiments; SF meta-analyzed published human data and provided critical intellectual input; DiK performed total body irradiation and designed bone marrow transfer experiments; and IP, MS, and GTS designed experiments, analyzed the data, and wrote the paper. GTS conceived the idea and supervised the study. All authors reviewed and concur with the submitted manuscript. All authors guarantee the study's integrity.

### Conflict of interest

The authors declare that they have no conflict of interest.

### For more information

Link to the Sanger Institute's Catalogue of Somatic Mutations In Cancer NRAS page: http://cancer.sanger.ac.uk/cosmic/gene/analysis?ln=NRAS

Link to homepage of the Laboratory where this work was carried out: http://www.lmrc.upatras.gr/

Link to microarray data reported in this study: http://www.ncbi.nlm.nih.gov/geo/query/acc.cgi?acc=GSE74309

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
