## [Review Process File · EMBO Molecular Medicine]

NRAS destines tumor cells to the lungs

Anastasios D. Giannou, Antonia arazioti, Nikolaos I. Kanellakis, Ioanna Giopanou, Ioannis Lilis, Dimitra E. Zazara, Giannoula Ntaliarda, Danai Kati, Vasileios Armenis, Georgia A. Giotopoulou, Anthi C. Krontira, Marina Lianou, Theodora galioti, Malamati Vreka, Maria Papageorgopoulou, Sotirios Fouzas, Dimitrios Kardamakis, Ioannis Psallidas, Magda Spella, and Georgios T. Stathopoulos

Corresponding author: Magda Spella and Georgios Stathopoulos, University of Patras

Review timeline:

Submission date:	22 August 2016
Editorial Decision:	23 September 2016
Revision received:	15 February 2017
Editorial Decision:	22 February 2017
Revision received:	24 February 2017
Accepted:	24 February 2017

Transaction Report:

Editor: Céline Carret

1st Editorial Decision

23 September 2016

Thank you for the submission of your manuscript to EMBO Molecular Medicine. We have now heard back from the two referees whom we asked to evaluate your manuscript. Although the referees find the study to be of potential interest, they also raise a number of concerns that need to be addressed in the next final version of your article.

As you will see from the comments pasted below, both referees are supportive of publication. While referee 2 has minor comments and suggests further clarifications here and there, referee 1 would like to see a better metastasis model *in vivo*, and additional experiments using shRNAs to provide controls and mechanism.

Given the balance of these evaluations, we feel that we can consider a revision of your manuscript if you can address the issues that have been raised within the time constraints outlined below. Please note that it is EMBO Molecular Medicine policy to allow only a single round of revision and that, as acceptance or rejection of the manuscript will depend on another round of review, your responses should be as complete as possible.

Please read below for important editorial formatting for submission of your revised article.

I look forward to receiving your revised manuscript.

***** Reviewer's comments *****

Referee #1 (Remarks):

Recent data point out that specific mutations in primary tumors could determine the site of metastatic invasion. This is new and very interesting concept in the metastasis biology that should be further explored. In this context, the results presented in this manuscript can help us better understand the possible association of N-Ras mutations in primary tumor with lung metastasis development. The data provided by the Authors is interesting and suggests that N-Ras mutants facilitate lung colonization by controlling the expression of IL-8 family chemokines, which impinge on the niche by modifying the lung endothelium and myeloid cell recruitment. The manuscript is concise, well written with solid and well presented data. However, there are some concerns that should be addressed.

Major points

1. My main concern is the use of lung tumor cell lines implanted subcutaneously to address the lung metastasis. This is an unlikely setting since contralateral lung mets are more typical. In addition, the prevalence of NRAS mutations in lung cancers is far less frequent than KRAS mutations and the lung cancer cell lines chosen carry both of these mutations. Thus, the central question is to what extent the effect detected is NRAS and not KRAS driven? In line with that, the authors only use one short hairpin RNA, and off target effects are a risk. Given that Ras genes present a significant sequence homology data regarding short-hairpin specificity is a must. Is expression of H-Ras or K-Ras affected by this short hairpin RNA? Additionally, a second independent shorthairpin against NRAS and not KRAS is needed at least for key experiments.

2. Based on the data from WBs presented in this manuscript the levels of NRas are significantly higher in AE17 and LLC cell lines comparing to MC38 cell line. Could it be that simply N-Ras overexpression and not N-Ras mutation is what actually facilitates lung colonization? Or a synergism between both effects drives the effect? The authors could overexpressing N-Ras wild type/mutant isoform in MC38 cell line and test which of these two conditions contributes to lung metastasis formation. What are the N-Ras levels in all cell lines presented in Figure 1A? Is there any difference in Ras downstream pathway activation between these cell lines?

Minor comments:

1. The sentence in abstract "In conclusion, mutant NRAS promotes the colonization of the lungs by various tumor types" seems an unnecessary overstatement as the mechanistic studies presented were obtained with mouse lung cancer cell lines. Similarly, in the discussion section there are unnecessary overstatements.

2. Previous evidence that the mutational status may influence the site of metastasis is shown in the literature, particularly in KRAS (i.e. Urosevic et al NCB 2014 and others). Please cite or modify the introduction accordingly (line 7 and 8 of 2 paragraph, page 3)

3. Please provide the data on primary tumor size for Figures 2A,B, C, D and 3F. Please explain how in 4 weeks metastatic bioluminescent signal of lungs could be stronger than that detected in primary tumor as shown in Figure 2A?

Referee #2 (Comments on Novelty/Model System):

The authors have used a variety of different human and mouse cells lines as well as several engineered animals models to validate their findings.

These findings have the potential to be further tested in clinical trials to reduce metastases in patients with Nras mutant tumors.

Referee #2 (Remarks):

In this study, Giannou and colleagues elegantly show that a NRASQ61K mutation promote lung metastasis by facilitating the extravasation of circulating tumor cells into the lung parenchyma. They go on and show that interleukin 8 related chemokine expression is responsible to initiate a communication between tumor cells, the pulmonary vasculature and myeloid cells. By suppressing chemokine signaling with Navarixin (a CXCR1/2 inhibitor) they were able to reduce lung colonies, suggesting that this therapy might prevent lung metastasis in patients with NRAS mutant primary tumors.

The authors provide a set of experiments that make an important contribution to the field of pulmonary metastasis and will be of broad interest. The experiments are very well designed and executed and the number of mice and the different strains used are impressive, significantly strengthening the conclusions of the manuscript.

Although no major revisions are needed, addressing the following comments and a more comprehensive presentation of the data (several times information within the text is missing) will strengthen the manuscript.

Minor comments:

1. Figure 1 misses the appropriate reference in the text when refereeing to the different panels. It will help the reader to follow each individual panel with a text reference. Some information could be moved to supplementary figures, ie: panel B is not needed in the main figure.

I am somehow puzzled that tumors were grown to almost 5cm³ when current legislation in Europe does not allow volume larger than 1.5 cm³. Was this needed to obtain metastasis?

2. Figure 2C. It is very difficult to see the lung figure on its current form. Are these confocal images? A higher magnification is suggested to better appreciate the stainings. At what time point were the pictures taken? I fail to see metastasis in these lungs.

It is not clear why MC38 cell line was chosen for the experiments in figure 2. Why not choosing a cell line that develops metastasis although at a lower rate without the Nras mutation? Like A549 or B16F10?

3. Figure 3A, B and C panels are confusing. What does it mean X MC38 on the Y axis on panel A? and % shC in panel B? A clearer way to represent the data of the QPCR would help the reader. The same will apply to figure 4. Further description on the quantification in the supplemental material should be provided.

Also not clear is panel C. The knock down in AE17 seems much better than in LLC, however the RNA data shows only 50% reduction in AE17 compared to 80% reduction in LLC. Figure 3E shows tumor growth s.c of about 1.5cm³ however figure 1C shows an average tumor volume of 4cm³ after 30 days for the same cell line. Is the discrepancy in the size responsible for the lower number of metastasis?

Figure 3G, MC38 was able to form forced metastasis when injected I.V. although with less efficiency than LLC and AE17. It is not clear why these numbers as well as HE pictures of the lungs were not included in figure 3G.

Figure 3H, could the authors comment why they choose to transfect HEK293T cells with a plasmid encoding NRASG61K and not any other of the cells lines presented in figure 1A, for which the mutational analysis was performed? In fact, MC38 was engineered to express pANRAS (see figure 4C) however the number and pictures of the lungs/metastasis was not provided.

4. They authors nicely show that NRAS mutation leads to the up-regulation of Cxcl5 (in mouse cell lines) or Cxcl6 (in human). I was wondering if the chemokine induction is the cause of metastatic colonization or the NRAS mutation per se. Could the authors show if overexpression of Cxcl5 or 6 in the absence of NRAS mutations leads to increase metastasis?

RESPONSE TO REVIEWERS' COMMENTS**Referee #1**

“Recent data point out that specific mutations in primary tumors could determine the site of metastatic invasion. This is new and very interesting concept in the metastasis biology that should be further explored. In this context, the results presented in this manuscript can help us better understand the possible association of N-Ras mutations in primary tumor with lung metastasis development. The data provided by the Authors is interesting and suggests that N-Ras mutants facilitate lung colonization by controlling the expression of IL-8 family chemokines, which impinge on the niche by modifying the lung endothelium and myeloid cell recruitment. The manuscript is concise, well written with solid and well presented data. However, there are some concerns that should be addressed.”

We sincerely thank this Reviewer for her/his positive overall appraisal of our work. We have undertaken every effort feasible to address her/his critiques.

“1. My main concern is the use of lung tumor cell lines implanted subcutaneously to address the lung metastasis. This is an unlikely setting since contralateral lung mets are more typical.”

We thank the Reviewer for the pertinent comment. We did not examine lung cancer cell lines only, but a variety of murine and human cell lines of pulmonary, pleural, skin, enteric, mammary, and other origins. This is why we chose the skin heterotopic model, which features automatic, spontaneous, physiologically relevant metastasis. This model is also used by leading authorities in the field (Acharyya et al., 2012; Chen et al., 2011; Lyden et al., 2001; Qian et al., 2011). We could not have used any orthotopic model to compare side-by-side the metastatic propensities of these cells, because this would bias the results, since the local microenvironment would be syngeneic to some tumor cell lines only and alien to others. This issue is now clarified in the revised introduction and results sections.

“In addition, the prevalence of NRAS mutations in lung cancers is far less frequent than KRAS mutations and the lung cancer cell lines chosen carry both of these mutations.”

Again we would like to stress that this study was not about lung cancer metastasis only, but was designed to mainly address lung-directed metastasis of cancer cells of any origin. Although NRAS mutations in lung cancer are infrequent, they do exist. Maybe it is these lung cancer that produce satellite nodules in the lung themselves? This will hopefully addressed by future clinical studies that are designed appropriately to tackle this question.

“Thus, the central question is to what extent the effect detected is NRAS and not KRAS driven? In line with that, the authors only use one short hairpin RNA, and off target effects are a risk. Given that Ras genes present a significant sequence homology data regarding short-hairpin specificity is a must. Is expression of H-Ras or K-Ras affected by this short hairpin RNA? Additionally, a second independent shorthairpin against NRAS and not KRAS is needed at least for key experiments.”

This is a key concern and we thank the Reviewer for pointing this out. We are sorry these issues were not clear enough in the initial submission. We used pools of three different shRNAs against NRAS, and not a single shRNA. These were provided as lentiviral pools by the vendor, so we could not use them separately. This would also increase the number of mice used for these studies, which is already very high (n = 611). To alleviate the Reviewer's valid concern, we now undertook new experiments using shRNA pools against *Kras* obtained from the same vendor and compared *Kras* and *Nras* silencing effects on primary tumor growth, pulmonary metastasis, and on NRAS and KRAS expression at the RNA levels in new Figure 4. In addition, we graphically provide the shRNA target sequences of the two different shRNA pools on the *Nras* and *Kras* transcripts in new Figure EV3. Our new results show that the shRNA pools used are specific and function to specifically silence the respective target transcripts without significant off-target effects on the undesired target transcript. In detail, *Nras* silencing decreased *Nras* but not *Kras* RNA levels in LLC and AE17 cells, while *Kras* silencing decreased *Kras* but not *Nras* RNA levels in LLC and AE17 cells (new Figure 4A). Moreover, that *Kras* signaling functions to promote both tumor growth and pulmonary metastasis, while *Nras* seems to more specifically promote lung metastasis, since it does not profoundly impact tumor growth, but is pivotal for pulmonary metastasis (new Figures 4C,

EV2H and EV2I). The Reviewer's concern, the relevant new experiments, and the revised interpretation of the data taking KRAS into account are now included in the revised manuscript.

"2. Based on the data from WBs presented in this manuscript the levels of NRAs are significantly higher in AE17 and LLC cell lines comparing to MC38 cell line. Could it be that simply N-Ras overexpression and not N-Ras mutation is what actually facilitates lung colonization? Or a synergism between both effects drives the effect? The authors could overexpressing N-Ras wild type/mutant isoform in MC38 cell line and test which of these two conditions contributes to lung metastasis formation."

The Reviewer is again correct, since RAS protein expression levels are of pivotal importance in addition to mutations (Stephen et al, 2014). To address this, we obtained a wild-type-NRAS encoding NRAS plasmid and overexpressed wild-type as well as Q61K-mutant NRAS in MC38 cells and injected stable clones s.c. The results, shown in new Figures 3B-3D and EV2F, indicate that either a mutation or elevated levels of wild-type NRAS are required for the observed phenotype of NRAS-mutant cells. The pertinent point of this Reviewer is also discussed in the revised Discussion section and changes were made throughout the manuscript, including title and abstract, to account for this point.

"What are the N-Ras levels in all cell lines presented in Figure 1A? Is there any difference in Ras downstream pathway activation between these cell lines?"

qPCR and immunoblot data of NRAS expression and of activation of RAS-downstream signaling pathways ERK, MAPK, NF- κ B, CREB, STAT, and NOTCH are now shown in new Figure EV1. The results indicate that, as this Reviewer and other authorities suspected, mutation is intrinsically associated with overexpression and overactivation of downstream signaling pathways. The new results are now discussed in the revised results and discussion sections.

"1. The sentence in abstract "In conclusion, mutant NRAS promotes the colonization of the lungs by various tumor types" seems an unnecessary overstatement as the mechanistic studies presented were obtained with mouse lung cancer cell lines. Similarly, in the discussion section there are unnecessary overstatements."

The sentence in question was rephrased to "In conclusion, NRAS promotes the colonization of the lungs by various tumor types in mouse models". Similar unnecessary overstatements in the discussion section were also amended or rephrased.

"2. Previous evidence that the mutational status may influence the site of metastasis is shown in the literature, particularly in KRAS (i.e. Urosevic et al NCB 2014 and others). Please cite or modify the introduction accordingly (line 7 and 8 of 2 paragraph, page 3)"

The relevant reference was included and the corresponding phrase was rephrased to "However, these signatures are hard to target, and direct links between a single cancer mutation and metastatic tropism to a given organ, such as those suggested by observational studies of NRAS and KRAS mutations in pulmonary and hepatic metastasis of colon cancer and melanoma (Lan et al, 2015; Urosevic et al, 2014; Tie et al, 2011; Ulivieri et al, 2015; Pereira et al, 2015), are intriguing and of potential clinical value".

"3. Please provide the data on primary tumor size for Figures 2 A, B, C, D and 3F. Please explain how in 4 weeks metastatic bioluminescent signal of lungs could be stronger than that detected in primary tumor as shown in Figure 2A?"

We are sorry to have mistakenly failed to identify in the legend and relevant text to Figures 2A and 2C that, for the purposes of these experiments, the primary tumors were either removed after two weeks and mice were allowed to survive (2A), as the primary signal largely overshadowed the metastatic signal, or were prematurely killed (2C). In the images shown in Figure 2A, the primary tumor signal stems from residual tumor tissue. This was corrected in the revised version of the manuscript. We are truly sorry for this mistake. Primary tumor size data for all main Figures are now included in new Figure EV2. However, we stress that, in all experiments except the ones shown in Figures 2A and 2C, mice were observed till moribund without intervention, in order to observe spontaneous, physiologically relevant metastasis.

Referee #2

“The authors have used a variety of different human and mouse cells lines as well as several engineered animals models to validate their findings. These findings have the potential to be further tested in clinical trials to reduce metastases in patients with Nras mutant tumors. In this study, Giannou and colleagues elegantly show that a NRASQ61K mutation promote lung metastasis by facilitating the extravasation of circulating tumor cells into the lung parenchyma. They go on and show that interleukin 8 related chemokine expression is responsible to initiate a communication between tumor cells, the pulmonary vasculature and myeloid cells. By suppressing chemokine signaling with Navarixin (a CXCR1/2 inhibitor) they were able to reduce lung colonies, suggesting that this therapy might prevent lung metastasis in patients with NRAS mutant primary tumors. The authors provide a set of experiments that make an important contribution to the field of pulmonary metastasis and will be of broad interest. The experiments are very well designed and executed and the number of mice and the different strains used are impressive, significantly strengthening the conclusions of the manuscript. Although no major revisions are needed, addressing the following comments and a more comprehensive presentation of the data (several times information within the text is missing) will strengthen the manuscript. “

We sincerely thank this Reviewer for her/his positive appraisal of our work. We really appreciate her/his constructive criticism and have undertaken every effort feasible to address her/his critiques.

“1. Figure 1 misses the appropriate reference in the text when refereeing to the different panels. It will help the reader to follow each individual panel with a text reference. Some information could be moved to supplementary figures, ie: panel B is not needed in the main figure.”

We sincerely apologize and corrected this reviewer’s pertinent comment. The description of Figure 1 in the results section was improved with appropriate callouts for each panel. Panel B and primary tumor growth data were moved to new Figures EV1A and EV1B to improve the legibility of Figure 1.

“I am somehow puzzled that tumors were grown to almost 5 cm³ when current legislation in Europe does not allow volume larger than 1.5 cm³. Was this needed to obtain metastasis?”

The Reviewer is again absolutely right. We obtained specific permits to let tumors grow out to 4-5 cm³ and to sacrifice animals only when moribund, because the phenomenon of spontaneous metastasis occurred after tumors grew to the EU limit of 1.5 cm³. Although waiting till these extreme tumor volumes were obtained was rarely necessary (we observed such tumor volumes only in 10 out of 611 mice used in these studies), this freedom to operate was necessary to conduct these studies since lung metastasis is a late manifestation of cancer. This is clarified in the revised Methods section and was properly addressed and approved in the animal care approvals obtained for these studies.

“2. Figure 2C. It is very difficult to see the lung figure on its current form. Are these confocal images? A higher magnification is suggested to better appreciate the stainings. At what time point were the pictures taken? I fail to see metastasis in these lungs.”

This reviewer is absolutely right. We sacrificed these animals prematurely (at two weeks post-tumor cell injections) and failed to state so in the text and the Figure legend of the initial submission. We are extremely sorry for this. The aim was to identify whether tumor and myeloid cells co-segregate in space in the pre-metastatic niche. Again we apologize and have corrected these aspects. These images were taken on a regular inverted microscope (Zeiss Axio Observed D1) not equipped with confocal capabilities. Confocal images of metastasis-seeded lungs with infiltrating myeloid cells expressing CXCR2 are shown in Figure 6D. The lung structure is not easily comprehended on such images as opposed to light microscopic images, even at higher magnification. We tried to improve the reader’s orientation by labeling some alveoli and bronchi in these images. We apologize to the Reviewer for the poor quality of these images, but ask for understanding, because these samples are two years old and have lost their intrinsic fluorescence, hence these images could not be replaced at this time.

“It is not clear why MC38 cell line was chosen for the experiments in figure 2. Why not choosing a cell line that develops metastasis although at a lower rate without the Nras mutation? Like A549 or B16F10?”

We thank the Reviewer for her/his sharp comment. The MC38 cell line was chosen because it is an adenocarcinoma that is syngeneic to the C57BL/6 mouse like LLC and AE17 cells and has a *Kras* mutation, alike the LLC and AE17 cells that feature both *Nras* and *Kras* mutations. A549 cells could not be used because they are human and need immunocompromized mice. B16F10 cells seed all organs with micrometastases (bullet-like pattern) without predilection but mice die from the primary tumor before the metastatic tumor cells can grow out (enhanced primary tumor growth rate compared to all other cell lines). In addition B16F10 cells do not have any of the mutations we looked for, including the *Kras* mutation, so they would be an incomplete control. In addition, we believe that MC38 cells are very hard controls for these studies, because they display aggressive tumor growth in the flank that is comparable to the two lung-metastatic lines as shown in new Figure EV2, and because they are *Kras*-mutant as the two lung-metastatic lines, so using them we controlled for *Kras* status.

“3. Figure 3A, B and C panels are confusing. What does it mean X MC38 on the Y axis on panel A? and % shC in panel B? A clearer way to represent the data of the QPCR would help the reader. The same will apply to figure 4. Further description on the quantification in the supplemental material should be provided.”

These issues, correctly pointed out by this Reviewer, were corrected throughout. x MC38 means times MC38 (was corrected and explained in the legend) and % shC means percent of corresponding control shRNA (was explained in the legend). qPCR quantification method details are now given in the Methods.

“Also not clear is panel C. The knock down in AE17 seems much better than in LLC, however the RNA data shows only 50% reduction in AE17 compared to 80% reduction in LLC.”

We cannot explain this discrepancy, but we are confident of the results shown, since experiments were redone three independent times. RNA and protein level silencing efficiencies were not consistent. However, we show beyond doubt in new Figure 4A and 4B that shRNA silenced NRAS and not KRAS at both the RNA and the protein level. One explanation for the divergent RNA and protein data would be different degrees of codon bias (J Clin Invest 2015;125:222–233, Current Biol 2013;23:70–75) in LLC and AE17 cells. For example, a greater scarcity of rare codons (which exist in NRAS) in AE17 cells would explain greater protein knock-down, even with less shRNA efficiency in terms of RNA silencing. We had no space to touch upon these issues in the revised manuscript, but would be happy to do so if the Reviewer wishes.

“Figure 3E shows tumor growth s.c of about 1.5cm3 however figure 1C shows an average tumor volume of 4 cm3 after 30 days for the same cell line. Is the discrepancy in the size responsible for the lower number of metastasis?”

Yes, exactly. We believe that this reduced tumor growth is due to the shRNA transfections and the *in vitro* puromycin treatment of the shRNA stable clones. Even immediately before injections *in vivo*, shRNA expressing stable clones necessitated low concentrations of puromycin in order for selective pressure to be maintained (1 µg/mL). In addition, flank tumor growth was not always the same for a given cell line and we observed significant variation over time, even taking the outmost care in order to exactly recapitulate experiments. This Reviewer is again right: the lower metastasis number is associated with a reduced primary tumor growth rate in this case. To give a comparative overview of all flank experiments, they were now grouped together in new Figure EV2.

“Figure 3G, MC38 was able to form forced metastasis when injected I.V. although with less efficiency than LLC and AE17. It is not clear why these numbers as well as HE pictures of the lungs were not included in figure 3G.”

These numbers and images were now included in the revised version (new Figure EV4D). MC38 can form forced metastases but to a lesser extent than *NRAS* mutant cells in our hands.

“Figure 3H, could the authors comment why they choose to transfect HEK293T cells with a plasmid encoding NRASG61K and not any other of the cells lines presented in figure 1A, for which the mutational analysis was performed? In fact, MC38 was engineered to express pANRAS (see figure 4C) however the number and pictures of the lungs/metastasis was not provided.”

We now show the data from mutant (and wild-type) NRAS-expressing MC38 cells in new Figure 3D. We also included wild-type NRAS plasmid-expressing MC38 cells, to see whether overexpression *per se* or mutation is necessary for metastasis, according to the criticism of another Reviewer. The results indicate that a mutation is required for the observed phenotype of NRAS-mutant cells, and not just elevated levels of wild-type NRAS. The HEK293T cells were chosen because they are not malignant and tumorigenic, but merely transformed. So we garnered that, if mutant NRAS can render even these cells spontaneously metastatic, this would be the hardest evidence ever that NRAS is a pro-metastasis gene. Indeed, as shown in new Figures 3E and EV2G, mutant NRAS-expressing HEK293T cells not only form late primary tumors (as do by the way control-transfected cells), but metastasize spontaneously to the lungs.

“4. The authors nicely show that NRAS mutation leads to the up-regulation of Cxcl5 (in mouse cell lines) or Cxcl6 (in human). I was wondering if the chemokine induction is the cause of metastatic colonization or the NRAS mutation *per se*. Could the authors show if overexpression of Cxcl5 or 6 in the absence of NRAS mutations leads to increase metastasis?”

We thank the Reviewer for this critical comment. To test this, we undertook additional *in vivo* experiments that unfortunately failed technically. Since we could not unequivocally show that CXCR1-cognate chemokines (in this case *Cxcl5*) promote pulmonary metastasis and are the NRAS-driven culprits of lung colonization, we address this in the revised discussion section.

2nd Editorial Decision

22 February 2017

Thank you for the submission of your revised manuscript to EMBO Molecular Medicine. We have now received the enclosed reports from the referees that were asked to re-assess it. As you will see the reviewers are now supportive and I am pleased to inform you that we will be able to accept your manuscript pending final editorial amendments.

Please submit your revised manuscript within two weeks.

I look forward to reading a new revised version of your manuscript as soon as possible.

***** Reviewer's comments *****

Referee #1 (Comments on Novelty/Model System):

The authors have addressed my main technical concern. Although indirectly, the evidence seem to confirm the suitability of their model.

Referee #1 (Remarks):

The authors have provided reasonable answers to most of the points raised. In particular and central to the previous limitations are the experiments focusing on the contribution of NRAS versus NRAS mutated. In other words, they clarify that it is the amount of NRAS and not just the mutation what drives the lung colonization phenotype. In addition, although no direct data is provided the experiments where NRAS is depleted while KRAS mutation is present indirectly clarify the contribution of each of the RAS isoforms. Overall, the manuscript has significantly improved through the revision process. Although, some remaining questions are still open, the clarifications to the most important points seem satisfactory and the work relevant for the community.

Referee #2 (Comments on Novelty/Model System):

The authors have included new experiments in the revised version to address the concerns raised by this reviewer. I am satisfied with the revised version.

Referee #2 (Remarks):

This reviewer is pleased to see the new data added by the authors, which substantially improved the overall message of the manuscript and is fully satisfied with the revised version submitted.

Corresponding Author Name: Georgios T. Stathopoulos

Manuscript Number: EMM-2016-06978-V3